# Inference for Convolutionally Observed Diffusion Processes

**DOI:** 10.3390/e22091031

**Published:** 2020-09-15

**Authors:** Shogo H Nakakita, Masayuki Uchida

**Affiliations:** 1Graduate School of Engineering Science, Osaka University, 1-3 Machikaneyamacho, Toyonaka, Osaka 560-0043, Japan; uchida@sigmath.es.osaka-u.ac.jp; 2Japanese Society for the Promotion of Science, 5 Chome-3-1 Kōjimachi, Chiyoda City, Tokyo 102-0083, Japan; 3The Ronin Institute for Independent Scholarship, 127 Haddon Place, Montclair, Montclair, NJ 07043, USA; 4Center for Mathematical Modeling and Data Science, Osaka University, 1-3 Machikaneyamacho, Toyonaka, Osaka 560-0043, Japan

**Keywords:** convolutional observation, diffusion processes, parametric inference, partial observation, statistical modelling, stochastic differential equations

## Abstract

We propose a new statistical observation scheme of diffusion processes named convolutional observation, where it is possible to deal with smoother observation than ordinary diffusion processes by considering convolution of diffusion processes and some kernel functions with respect to time parameter. We discuss the estimation and test theories for the parameter determining the smoothness of the observation, as well as the least-square-type estimation for the parameters in the diffusion coefficient and the drift one of the latent diffusion process. In addition to the theoretical discussion, we also examine the performance of the estimation and the test with computational simulation, and show an example of real data analysis for one EEG data whose observation can be regarded as smoother one than ordinary diffusion processes with statistical significance.

## 1. Introduction

We consider a *d*-dimensional diffusion process defined by the following stochastic differential equation,
$$\begin{array}{c} dX_{t}=b\left(X_{t},\beta \right)dt+a\left(X_{t},\alpha \right)dw_{t},\;X_{-\lambda }=x_{-\lambda }, \end{array}$$
where λ>0, ${\left\{w_{t}\right\}}_{t\geq -\lambda }$ is a standard *r*-dimensional Wiener process, $x_{-\lambda }$ is an $R^{d}$-valued random variable independent of ${\left\{w_{t}\right\}}_{t\geq -\lambda }$, $\alpha \in \Theta_{1}$ and $\beta \in \Theta_{2}$ are unknown parameters, $\Theta_{1}\subset R^{m_{1}}$ and $\Theta_{2}\subset R^{m_{2}}$ are compact and convex parameter spaces, $a:R^{d}\times \Theta_{1}\to R^{d}⊗R^{r}$ and $b:R^{d}\times \Theta_{2}\to R^{d}$ are known functions. Our concern is statistical estimation for α and β from observation. $\theta_{⋆}=\left(\alpha_{⋆},\beta_{⋆}\right)$ denotes the true value of $\theta :=\left(\alpha ,\beta \right)$.

We denote the observation as the discretised process $\left\{{\bar{X}}_{ih_{n},n}:i=0,\ldots ,n\right\}$ with discretisation step $h_{n}>0$ such that $h_{n}\to 0$ and $T_{n}:=nh_{n}\to \infty$, where the convoluted process ${\left\{{\bar{X}}_{t,n}\right\}}_{t\geq 0}$ is defined as
$$\begin{array}{c} {\bar{X}}_{t,n}:=\int_{t-\bar{\rho }h_{n}}^{t}V_{h_{n}}\left(t-s\right)X_{s}ds=\int_{R}V_{h_{n}}\left(t-s\right)X_{s}ds=\left(V_{h_{n}}*X\right)\left(t\right), \end{array}$$
where $V_{h_{n}}$ is an $R^{d}⊗R^{d}$-valued kernel function whose support is a subset of $\left[0,\bar{\rho }h_{n}\right]$, and $\bar{\rho }>0$ such that $\sup_{n}\bar{\rho }h_{n}\leq \lambda$. In this paper, we specify $V_{h_{n}}=V_{\rho ,h_{n}}$ which is a parametric kernel function whose support is a subset of $\left[0,\bar{\rho }h_{n}\right]$ defined as
$$\begin{array}{c} V_{\rho ,h_{n}}^{\left(i,j\right)}\left(t\right):=\left\{\begin{array}{cc} {\left(\rho^{\left(i\right)}h_{n}\right)}^{-1}1_{\left[0,\rho^{\left(i\right)}h_{n}\right]}\left(t\right) & \mathrm{if}\;i=j\;\mathrm{and}\;\rho^{\left(i\right)}>0,\\ \delta \left(t\right) & \mathrm{if}\;i=j\;\mathrm{and}\;\rho^{\left(i\right)}=0,\\ 0 & \mathrm{if}\;i\neq j, \end{array}\right. \end{array}$$
$\delta \left(t\right)$ is the Dirac-delta function, $\rho ={\left[\rho^{\left(1\right)},\ldots ,\rho^{\left(d\right)}\right]}^{T}\in \Theta_{\rho }:={\left[0,\bar{\rho }\right]}^{d}$ is the smoothing parameter determining the smoothness of observation. That is to say, the observed process is defined as follows:$$\begin{array}{c} {\bar{X}}_{ih_{n},n}^{\left(\ell \right)}=\left\{\begin{array}{cc} {\left(\rho^{\left(\ell \right)}h_{n}\right)}^{-1}\int_{\left(i-\rho^{\left(\ell \right)}\right)h_{n}}^{ih_{n}}X_{s}^{\left(\ell \right)}ds & \;\mathrm{if}\;\rho^{\left(\ell \right)}>0,\\ X_{ih_{n}}^{\left(\ell \right)} & \;\mathrm{if}\;\rho^{\left(\ell \right)}=0, \end{array}\right. \end{array}$$
for all ℓ=1,…,d. Let us consider both the problems that (i) ρ is a known parameter, and (ii) ρ is an unknown one whose true value is denoted by $\rho_{⋆}$ and this is estimated by observation $\left\{{\bar{X}}_{ih_{n},n}\right\}$, and the parameter space is denoted as $\Xi :=\Theta_{\rho }\times \Theta$, where $\Theta :=\Theta_{1}\times \Theta_{2}$.

When assuming ρ as a known parameter, we can find literature for parametric estimation for α and/or β based on observation schemes which can be represented as special cases for some specific ρ. If ρ=0, our scheme is simply equivalent to parametric inference based on discretely observed diffusion processes $\left\{X_{ih_{n}}:i=0,\ldots ,n\right\}$ studied in [1,2,3,4,5,6,7,8] and references therein. If $\rho ={\left[1,\ldots ,1\right]}^{T}$, we can regard the problem as parametric estimation for integrated diffusion processes discussed in Gloter [9], Ditlevsen and Sørensen [10], Gloter [11], Gloter and Gobet [12], Sørensen [13]. Even for the case $\rho ={\left[0,\ldots ,0,1,\ldots ,1\right]}^{T}$ where some axes correspond to direct observation and the others do to integrated observation, we give consistent estimators for α and β by considering the scheme of convolutionally observed diffusion processes and this is one of the contributions of our study.

What is more, our contribution is to consider the scheme where ρ is unknown and succeed in representation of the microstructure noise which makes the observation smoother than the latent diffusion process itself, which can be seen in some biomedical time series data. Statistical modelling of biomedical time series data with stochastic differential equations has been one of the topics eagerly studied e.g., [14,15,16,17]. As [18] states, the existence of microstructure noise in financial data affects realised volatilities to increase as the subsampling frequency gets higher, for instance, see Figure 7.1 on p. 217 in [19]. However, realised volatilities of some biomedical data such as EEG decrease as subsampling frequency increases. For instance, some time series data for the 2nd participant in the dataset named Two class motor imagery (002-2014) of BNCI Horizon 2020 [20] show clear tendency of decreasing realised volatilities as subsampling frequency increases. Figure 1 shows the path of the 2nd axis of the data S02E.mat BNCI Horizon 2020 [20] for all 222 seconds (the observation frequency is 512 Hz, and hence the entire data size is 113664) and that for the first one second; it seems to perturb like a diffusion process.

We define realised volatilities with subsampling as for a sequence of real-valued observation ${\left\{Y_{i}\right\}}_{i=0,\ldots ,n}$,
$$\begin{array}{c} RV\left(k\right)=\sum_{1\leq i\leq \left[n/k\right]}{\left(Y_{ik}-Y_{\left(i-1\right)k}\right)}^{2}, \end{array}$$
where k=1,…,100 is the subsampling frequency parameter, and provide a plot of the realised volatilities the 2nd axis of the data S02E.mat in Figure 2:

The altitudes of the graph represented in the *y*-axis correspond to the values of the realised volatilities $RV\left(k\right)$ with subsampling at every *k* observation represented in the *x*-axis. It is observable that the increasing subsampling frequency results in decreasing realised volatilities, which cannot be explained by the existent major microstructure noises, e.g., see [21,22,23,24,25]. To explain this phenomenon, we consider the smoother process than the latent one, though ordinarily microstructure noises make the observation rougher than the latent process, because quadratic variation of a sufficiently smooth function is zero. One way to deal with smoother observation than the latent state is convolutional observation. As a concrete example, we show a convolutionally observed diffusion process and its characteristics in realised volatilities: let us consider the following 1-dimensional stochastic differential equation defining an Ornstein–Uhlenbeck (OU) process:$$\begin{array}{c} dX_{t}=-20X_{t}dt+10dw_{t},X_{-\lambda }=0, \end{array}$$
where $\lambda =10^{-2/5}$. We simulate the stochastic differential equation by Euler–Maruyama method see [26] with parameters $n=10^{7}$, $h_{n}=10^{-5}$, and $T_{n}=10^{2}$ and its convolution approximated by summation with the smoothing parameter ρ=10 (for details, see Section 5). Figure 3 shows the latent diffusion process and the convoluted observation on $\left[0,1\right]$, and we can see that the observation is indeed smoothed compared to the latent state.

In Figure 4, we also give the plot of realised volatilities of the convolutionally observed process with subsampling as Figure 2.

It is seen that the convolutional observation of a diffusion process also has the characteristics of decreasing realised volatilities as subsampling frequency increases, which can bee seen in some biological data such as BNCI Horizon 2020 [20]. Of course, graphically comparing characteristics of simulation and real data is insufficient to verify the convolutional observation with smoothing parameter ρ>0 in 1-dimensional case; therefore, we propose statistical estimation method for unknown ρ and hypothesis test with the null hypothesis $H_{0}:$ρ=0 and the alternative one $H_{1}:$ρ≠0 from convolutional observation in Section 3. Moreover, in Section 6, we examine the real EEG data plotted in Figure 2 by the statistical hypothesis testing we propose, and see it is more appropriate to consider the data as a convolutional observation of a latent diffusion process with ρ≠0 rather than direct observation of the latent process, which indicates the validity to deal with the problem of the convolutional observation scheme with unknown ρ.

The paper is composed of the following sections: Section 2 gives the notations and assumptions used in this paper; Section 3 discusses the estimation and test for smoothing parameter ρ, and the discussion provides us with the tools to examine whether we should consider the convolutional observation scheme; Section 4 proposes the quasi-likelihood functions for the parameter of diffusion processes α and β, and corresponding estimators with consistency; Section 5 is for the computational simulation to examine the theoretical results in the previous sections; and Section 6 shows an application of the methods we propose in real data analysis.

## 2. Notations and Assumptions

### 2.1. Notations

First of all, we set $A\left(x,\alpha \right):=a{\left(x,\alpha \right)}^{⊗2}$, $a\left(x\right):=a\left(x,\alpha_{⋆}\right)$, $A\left(x\right):=A\left(x,\alpha_{⋆}\right)$ and $b\left(x\right):=b\left(x,\beta_{⋆}\right)$. We also give the notation for a matrix-valued function $G\left(x,\alpha |\rho \right)$ such that $G^{\left(i,j\right)}\left(x,\alpha |\rho \right):=A^{\left(i,j\right)}\left(x,\alpha \right)f_{G}\left(\rho^{\left(i\right)},\rho^{\left(j\right)}\right)$, where
$$\begin{array}{cc} \; & f_{G}\left(\rho^{\left(i\right)},\rho^{\left(j\right)}\right)\\ \; & :=\left\{\begin{array}{cc} 1 & \;\mathrm{if}\;\rho^{\left(i\right)}=\rho^{\left(j\right)}=0,\\ 1-\frac{\rho^{\left(j\right)}}{2} & \;\mathrm{if}\;\rho^{\left(i\right)}=0,\rho^{\left(j\right)}\in \left(0,1\right],\\ \frac{1}{2\rho^{\left(j\right)}} & \;\mathrm{if}\;\rho^{\left(i\right)}=0,\rho^{\left(j\right)}\in \left(1,\bar{\rho }\right],\\ 1-\frac{\rho^{\left(i\right)}}{2} & \;\mathrm{if}\;\rho^{\left(i\right)}\in \left(0,1\right],\rho^{\left(j\right)}=0,\\ \frac{1}{2\rho^{\left(i\right)}} & \;\mathrm{if}\;\rho^{\left(i\right)}\in \left(1,\bar{\rho }\right],\rho^{\left(j\right)}=0,\\ \frac{-3{\left(\rho^{\left(i\right)}\right)}^{2}\rho^{\left(j\right)}+3\rho^{\left(i\right)}{\left(\rho^{\left(j\right)}\right)}^{2}+6\rho^{\left(i\right)}\rho^{\left(j\right)}-2{\left(\rho^{\left(j\right)}\right)}^{3}}{6\rho^{\left(i\right)}\rho^{\left(j\right)}} & \;\mathrm{if}\;\rho^{\left(i\right)},\rho^{\left(j\right)}\in \left(0,1\right],\rho^{\left(i\right)}>\rho^{\left(j\right)},\\ \frac{3{\left(\rho^{\left(i\right)}\right)}^{2}\rho^{\left(j\right)}-3\rho^{\left(i\right)}{\left(\rho^{\left(j\right)}\right)}^{2}+6\rho^{\left(i\right)}\rho^{\left(j\right)}-2{\left(\rho^{\left(i\right)}\right)}^{3}}{6\rho^{\left(i\right)}\rho^{\left(j\right)}} & \;\mathrm{if}\;\rho^{\left(i\right)},\rho^{\left(j\right)}\in \left(0,1\right],\rho^{\left(i\right)}\leq \rho^{\left(j\right)},\\ \frac{3{\left(\rho^{\left(j\right)}\right)}^{2}+3\rho^{\left(j\right)}-{\left(\rho^{\left(j\right)}\right)}^{3}}{6\rho^{\left(i\right)}\rho^{\left(j\right)}} & \;\mathrm{if}\;\rho^{\left(i\right)}\in \left(1,\bar{\rho }\right],\rho^{\left(j\right)}\in \left(0,1\right],\rho^{\left(i\right)}>\rho^{\left(j\right)}+1,\\ \frac{6\rho^{\left(j\right)}-1}{6\rho^{\left(i\right)}\rho^{\left(j\right)}} & \;\mathrm{if}\;\rho^{\left(i\right)},\rho^{\left(j\right)}\in \left(1,\bar{\rho }\right],\rho^{\left(i\right)}>\rho^{\left(j\right)}+1,\\ \frac{{\left(\rho^{\left(i\right)}-\rho^{\left(j\right)}\right)}^{3}-3{\left(\rho^{\left(i\right)}\right)}^{2}+6\rho^{\left(i\right)}\rho^{\left(j\right)}+3\rho^{\left(i\right)}-1-{\left(\rho^{\left(j\right)}\right)}^{3}}{6\rho^{\left(i\right)}\rho^{\left(j\right)}} & \;\mathrm{if}\;\rho^{\left(i\right)}\in \left(1,\bar{\rho }\right],\rho^{\left(j\right)}\in \left(0,1\right],\rho^{\left(i\right)}\leq \rho^{\left(j\right)}+1,\\ \frac{{\left(\rho^{\left(i\right)}-\rho^{\left(j\right)}\right)}^{3}-3{\left(\rho^{\left(i\right)}\right)}^{2}+6\rho^{\left(i\right)}\rho^{\left(j\right)}+3\rho^{\left(i\right)}-3{\left(\rho^{\left(j\right)}\right)}^{2}+3\rho^{\left(j\right)}-2}{6\rho^{\left(i\right)}\rho^{\left(j\right)}} & \;\mathrm{if}\;\rho^{\left(i\right)},\rho^{\left(j\right)}\in \left(1,\bar{\rho }\right],\rho^{\left(j\right)}<\rho^{\left(i\right)}\leq \rho^{\left(j\right)}+1,\\ \frac{-{\left(\rho^{\left(i\right)}-\rho^{\left(j\right)}\right)}^{3}+6\rho^{\left(i\right)}\rho^{\left(j\right)}-{\left(\rho^{\left(i\right)}\right)}^{3}-3{\left(\rho^{\left(j\right)}\right)}^{2}+3\rho^{\left(j\right)}-1}{6\rho^{\left(i\right)}\rho^{\left(j\right)}} & \;\mathrm{if}\;\rho^{\left(i\right)}\in \left(0,1\right],\rho^{\left(j\right)}\in \left(1,\bar{\rho }\right],\rho^{\left(j\right)}\leq \rho^{\left(i\right)}+1,\\ \frac{-{\left(\rho^{\left(i\right)}-\rho^{\left(j\right)}\right)}^{3}-3{\left(\rho^{\left(i\right)}\right)}^{2}-3{\left(\rho^{\left(j\right)}\right)}^{2}+6\rho^{\left(i\right)}\rho^{\left(j\right)}+3\rho^{\left(i\right)}+3\rho^{\left(j\right)}-2}{6\rho^{\left(i\right)}\rho^{\left(j\right)}} & \;\mathrm{if}\;\rho^{\left(i\right)},\rho^{\left(j\right)}\in \left(1,\bar{\rho }\right],\rho^{\left(i\right)}\leq \rho^{\left(j\right)}\leq \rho^{\left(i\right)}+1,\\ \frac{3{\left(\rho^{\left(i\right)}\right)}^{2}+3\rho^{\left(i\right)}-{\left(\rho^{\left(i\right)}\right)}^{3}}{6\rho^{\left(i\right)}\rho^{\left(j\right)}} & \;\mathrm{if}\;\rho^{\left(i\right)}\in \left(0,1\right],\rho^{\left(j\right)}\in \left(1,\bar{\rho }\right],\rho^{\left(j\right)}>\rho^{\left(i\right)}+1,\\ \frac{6\rho^{\left(i\right)}-1}{6\rho^{\left(i\right)}\rho^{\left(j\right)}} & \;\mathrm{if}\;\rho^{\left(i\right)},\rho^{\left(j\right)}\in \left(1,\bar{\rho }\right],\rho^{\left(j\right)}>\rho^{\left(i\right)}+1. \end{array}\right. \end{array}$$

The continuity of the function $f_{G}$ is shown in the Appendix A. For the detailed discussion of the necessity of $f_{G}$, see Remark 4.

In addition, we also give the notation used throughout this paper.
For every matrix *A*, $A^{T}$ is the transpose of *A*, and $A^{⊗2}:=AA^{T}$.For every set of matrices *A* and *B* of the same size, $A\left[B\right]:=\mathrm{tr}\left(AB^{T}\right)$. Moreover, for any m∈N, $A\in R^{m}⊗R^{m}$ and $u,v\in R^{m}$, $A\left[u,v\right]:=v^{T}Au$.$v^{\left(\ell \right)}$ and $A^{\left(\ell_{1},\ell_{2}\right)}$ denote the *ℓ*-th element of a vector *v* and the $\left(\ell_{1},\ell_{2}\right)$-th one of a matrix *A*, respectively.For any vector *v* and any matrix *A*, $\left|v\right|:=\sqrt{\mathrm{tr}\left(v^{T}v\right)}$ and $∥A∥:=\sqrt{\mathrm{tr}\left(A^{T}A\right)}$.$\left(\Omega ,P,F,F_{t}\right)$ denotes the stochastic basis, where $F_{t}:=\sigma \left(x_{-\lambda },w_{s}:s\leq t\right)$.$\mu \left(f\left(\cdot \right)\right)$ denotes the integral of a μ-integrable function *f* where μ is a measure.

### 2.2. Assumptions

With respect to $X_{t}$, we assume the following conditions.

[A1]
(i)For a constant *C*, for all $x_{1},x_{2}\in R^{d}$,
$$\begin{array}{c} \underset{\alpha \in \Theta_{1}}{\sup}∥a\left(x_{1},\alpha \right)-a\left(x_{2},\alpha \right)∥+\underset{\beta \in \Theta_{2}}{\sup}\left|b\left(x_{1},\beta \right)-b\left(x_{2},\beta \right)\right|\leq C\left|x_{1}-x_{2}\right|. \end{array}$$(ii)For all p≥0, $\sup_{t\geq -\lambda }E_{\theta_{⋆}}\left[{\left|X_{t}\right|}^{p}\right]<\infty$.(iii)There exists a unique invariant measure $\nu_{0}$ on $\left(R^{d},B\left(R^{d}\right)\right)$ and for all p≥1 and $f\in L^{p}\left(\nu_{0}\right)$ with polynomial growth,
$$\begin{array}{c} \frac{1}{T}\int_{-\lambda }^{T}f\left(X_{t}\right)dt\to^{P}\int_{R^{d}}f\left(x\right)\nu_{0}\left(dx\right). \end{array}$$
**Remark** **1.**
*For sufficient conditions of these regularity ones, please see [A1] and Remark 1 in Uchida and Yoshida [7].*

[A2]There exists C>0 such that $a:R^{d}\times \Theta_{1}\to R^{d}⊗R^{r}$ and $b:R^{d}\times \Theta_{2}\to R^{d}$ have continuous derivatives satisfying
$$\begin{array}{cc} \underset{\alpha \in \Theta_{1}}{\sup}\left|\partial_{x}^{j}\partial_{\alpha }^{i}a\left(x,\alpha \right)\right| & \leq C{\left(1+\left|x\right|\right)}^{C},\;0\leq i\leq 2,\;0\leq j\leq 2,\\ \underset{\beta \in \Theta_{2}}{\sup}\left|\partial_{x}^{j}\partial_{\beta }^{i}b\left(x,\beta \right)\right| & \leq C{\left(1+\left|x\right|\right)}^{C},\;0\leq i\leq 2,\;0\leq j\leq 2. \end{array}$$With the invariant measure $\nu_{0}$, $\xi :={\left[\rho^{T},\theta^{T}\right]}^{T}$, and $\xi_{⋆}$ denoting the true value of ξ, we define
$$\begin{array}{cc} V_{1}\left(\alpha |\xi_{⋆}\right)\; & :=-\int_{R^{d}}{∥G\left(x,\alpha |\rho_{⋆}\right)-G\left(x,\alpha_{⋆}|\rho_{⋆}\right)∥}^{2}\nu_{0}\left(dx\right),\\ V_{2}\left(\beta |\xi_{⋆}\right)\; & :=-\int_{R^{d}}{\left|b\left(x,\beta \right)-b\left(x,\beta_{⋆}\right)\right|}^{2}\nu_{0}\left(dx\right). \end{array}$$For these functions, let us assume the following identifiability conditions hold.[A3]There exist $\chi_{1}\left(\alpha_{⋆}\right)>0$ and $\chi_{1}^{\prime }\left(\beta_{⋆}\right)>0$ such that for all $\alpha \in \Theta_{1}$ and $\beta \in \Theta_{2}$, $V_{1}\left(\alpha |\xi_{⋆}\right)\leq -\chi_{1}\left(\alpha_{⋆}\right){\left|\alpha -\alpha_{⋆}\right|}^{2}$ and $V_{2}\left(\beta |\xi_{⋆}\right)\leq -\chi_{1}^{\prime }\left(\beta_{⋆}\right){\left|\beta -\beta_{⋆}\right|}^{2}$.

## 3. Estimation and Test of the Smoothing Parameter

In this section, we discuss the case where the smoothing parameter ρ of the kernel function $V_{\rho ,h_{n}}$ is unknown. The estimation is significant for estimation of α and β since we utilise the estimate for ρ in quasi-likelihood functions of α and β. The test problem for hypotheses $H_{0}:\rho =0$ and $H_{1}:\rho \neq 0$ is also important to examine whether our framework of convolutional observation is meaningful.

### 3.1. Estimation of the Smoothing Parameter

For simplicity of notation, let us consider the case $\bar{\rho }>2$; otherwise the discussion is quite parallel. We should note that for all i=1,…,d,
$$\begin{array}{c} G^{\left(i,i\right)}\left(x|\rho \right)=\left\{\begin{array}{cc} A^{\left(i,i\right)}\left(x\right) & \;\mathrm{if}\;\rho^{\left(i\right)}=0,\\ A^{\left(i,i\right)}\left(x\right)\left(1-\frac{\rho^{\left(i\right)}}{3}\right) & \;\mathrm{if}\;\rho^{\left(i\right)}\in \left(0,1\right],\\ A^{\left(i,i\right)}\left(x\right)\left(\frac{1}{\rho^{\left(i\right)}}-\frac{1}{3{\left(\rho^{\left(i\right)}\right)}^{2}}\right) & \;\mathrm{if}\;\rho^{\left(i\right)}\in \left(1,\bar{\rho }\right]. \end{array}\right. \end{array}$$

Let us consider the estimation of $\rho^{\left(i\right)}$ with using the next statistics: the full quadratic variation
$$\begin{array}{c} \frac{1}{nh_{n}}\sum_{k=1}^{n}{\left({\bar{X}}_{kh_{n},n}^{\left(i\right)}-{\bar{X}}_{\left(k-1\right)h_{n},n}^{\left(i\right)}\right)}^{2}\to^{P}\left\{\begin{array}{cc} \nu_{0}\left(A^{\left(i,i\right)}\left(\cdot \right)\right) & \;\mathrm{if}\;\rho_{⋆}^{\left(i\right)}=0,\\ \nu_{0}\left(A^{\left(i,i\right)}\left(\cdot \right)\right)\left(1-\frac{\rho_{⋆}^{\left(i\right)}}{3}\right) & \;\mathrm{if}\;\rho_{⋆}^{\left(i\right)}\in \left(0,1\right],\\ \nu_{0}\left(A^{\left(i,i\right)}\left(\cdot \right)\right)\left(\frac{1}{\rho_{⋆}^{\left(i\right)}}-\frac{1}{3{\left(\rho_{⋆}^{\left(i\right)}\right)}^{2}}\right) & \;\mathrm{if}\;\rho_{⋆}^{\left(i\right)}\in \left(1,\bar{\rho }\right], \end{array}\right. \end{array}$$
because of Proposition 3 in Appendix A, and the reduced quadratic variation defined as $\frac{1}{nh_{n}}\sum_{2\leq 2k\leq n}{\left({\bar{X}}_{2kh_{n},n}^{\left(i\right)}-{\bar{X}}_{\left(2k-2\right)h_{n},n}^{\left(i\right)}\right)}^{2}$ converges in probability as follows.

**Lemma** **1.**
*Under [A1], we have the convergence in probability such that*
$$\begin{array}{cc} \; & \frac{1}{nh_{n}}\sum_{2\leq 2k\leq n}{\left({\bar{X}}_{2kh_{n},n}^{\left(i\right)}-{\bar{X}}_{\left(2k-2\right)h_{n},n}^{\left(i\right)}\right)}^{2}\\ \; & \to^{P}\left\{\begin{array}{cc} \nu_{0}\left(A^{\left(i,i\right)}\left(\cdot \right)\right) & \;\mathrm{if}\;\rho_{⋆}^{\left(i\right)}=0,\\ \nu_{0}\left(A^{\left(i,i\right)}\left(\cdot \right)\right)\left(1-\frac{\rho_{⋆}^{\left(i\right)}}{6}\right) & \;\mathrm{if}\;\rho_{⋆}^{\left(i\right)}\in \left(0,2\right],\\ \nu_{0}\left(A^{\left(i,i\right)}\left(\cdot \right)\right)\left(\frac{2}{\rho_{⋆}^{\left(i\right)}}-\frac{4}{3{\left(\rho_{⋆}^{\left(i\right)}\right)}^{2}}\right) & \;\mathrm{if}\;\rho_{⋆}^{\left(i\right)}\in \left(2,\bar{\rho }\right]. \end{array}\right. \end{array}$$


Then we define the ratio of those statistics such that
$$\begin{array}{cc} R_{n}^{\left(i\right)} & :=\left(\frac{1}{nh_{n}}\sum_{k=1}^{n}{\left({\bar{X}}_{kh_{n},n}^{\left(i\right)}-{\bar{X}}_{\left(k-1\right)h_{n},n}^{\left(i\right)}\right)}^{2}\right){\left(\frac{1}{nh_{n}}\sum_{2\leq 2k\leq n}{\left({\bar{X}}_{2kh_{n},n}^{\left(i\right)}-{\bar{X}}_{\left(2k-2\right)h_{n},n}^{\left(i\right)}\right)}^{2}\right)}^{-1}\\ & \to^{P}\left\{\begin{array}{cc} 1 & \;\mathrm{if}\;\rho_{⋆}^{\left(i\right)}=0,\\ \left(1-\frac{\rho_{⋆}^{\left(i\right)}}{3}\right){\left(1-\frac{\rho_{⋆}^{\left(i\right)}}{6}\right)}^{-1} & \;\mathrm{if}\;\rho_{⋆}^{\left(i\right)}\in \left(0,1\right],\\ \left(\frac{1}{\rho_{⋆}^{\left(i\right)}}-\frac{1}{3{\left(\rho_{⋆}^{\left(i\right)}\right)}^{2}}\right){\left(1-\frac{\rho_{⋆}^{\left(i\right)}}{6}\right)}^{-1} & \;\mathrm{if}\;\rho_{⋆}^{\left(i\right)}\in \left(1,2\right],\\ \left(\frac{1}{\rho_{⋆}^{\left(i\right)}}-\frac{1}{3{\left(\rho_{⋆}^{\left(i\right)}\right)}^{2}}\right){\left(\frac{2}{\rho_{⋆}^{\left(i\right)}}-\frac{4}{3{\left(\rho_{⋆}^{\left(i\right)}\right)}^{2}}\right)}^{-1} & \;\mathrm{if}\;\rho_{⋆}^{\left(i\right)}\in \left(2,\bar{\rho }\right] \end{array}\right.\\ & =\left\{\begin{array}{cc} 1 & \;\mathrm{if}\;\rho^{\left(i\right)}=0,\\ \left(6-2\rho_{⋆}^{\left(i\right)}\right){\left(6-\rho_{⋆}^{\left(i\right)}\right)}^{-1} & \;\mathrm{if}\;\rho_{⋆}^{\left(i\right)}\in \left(0,1\right],\\ \left(6\rho_{⋆}^{\left(i\right)}-2\right){\left(6{\left(\rho_{⋆}^{\left(i\right)}\right)}^{2}-{\left(\rho_{⋆}^{\left(i\right)}\right)}^{3}\right)}^{-1} & \;\mathrm{if}\;\rho_{⋆}^{\left(i\right)}\in \left(1,2\right],\\ \left(3\rho_{⋆}^{\left(i\right)}-1\right){\left(6\rho_{⋆}^{\left(i\right)}-4\right)}^{-1} & \;\mathrm{if}\;\rho_{⋆}^{\left(i\right)}\in \left(2,\bar{\rho }\right], \end{array}\right.\\ & =:R\left(\rho_{⋆}^{\left(i\right)}\right), \end{array}$$
where *R* has the next property.

**Lemma** **2.**
*R is a $\left[\left(3\bar{\rho }-1\right){\left(6\bar{\rho }-4\right)}^{-1},1\right]$-valued monotonically decreasing continuous function, and has a continuous inverse $R^{-1}:\left[\left(3\bar{\rho }-1\right){\left(6\bar{\rho }-4\right)}^{-1},1\right]\to \left[0,\bar{\rho }\right]$.*


We define ${\hat{\rho }}_{n}$ such that
$$\begin{array}{cc} {\hat{\rho }}_{n}^{\left(i\right)} & :=\left\{\begin{array}{cc} 0 & \;\mathrm{if}\;R_{n}^{\left(i\right)}>1,\\ R^{-1}\left(R_{n}^{\left(i\right)}\right) & \;\mathrm{if}\;R_{n}^{\left(i\right)}\in \left[\left(3\bar{\rho }-1\right){\left(6\bar{\rho }-4\right)}^{-1},1\right],\\ \bar{\rho } & \;\mathrm{if}\;R_{n}^{\left(i\right)}<\left(3\bar{\rho }-1\right){\left(6\bar{\rho }-4\right)}^{-1}, \end{array}\right. \end{array}$$
and then continuous mapping theorem for convergence in probability verifies the next result.

**Theorem** **1.**
*Under [A1], ${\hat{\rho }}_{n}$ has consistency, i.e., ${\hat{\rho }}_{n}\to^{P}\rho_{⋆}$.*


**Remark** **2.**
*We can compute $y=R^{-1}\left(x\right),\;x\in \left[\left(3\bar{\rho }-1\right){\left(6\bar{\rho }-4\right)}^{-1},1\right]$ by solving the following equations:*
$$\begin{array}{cccc} \left(i\right)\; & x=\left(6-2y\right){\left(6-y\right)}^{-1} & \; & \;\mathrm{if}\;x\in \left(4/5,1\right],\\ \left(ii\right)\; & x=\left(6-2y\right){\left(6y^{2}-y^{3}\right)}^{-1} & \; & \;\mathrm{if}\;x\in \left(5/8,4/5\right],\\ \left(iii\right)\; & x=\left(3y-1\right){\left(6y-4\right)}^{-1} & \; & \;\mathrm{if}\;x\in \left[\left(3\bar{\rho }-1\right){\left(6\bar{\rho }-4\right)}^{-1},5/8\right]. \end{array}$$


### 3.2. Test for Smoothed Observation

For all i=1,…,d, we consider the next hypothesis testing:$$\begin{array}{c} H_{0}:\rho^{\left(i\right)}=0,\;H_{1}:\rho^{\left(i\right)}>0. \end{array}$$
Let us consider the following test statistic:$$\begin{array}{cc} T_{i,n} & :=\sqrt{\frac{n}{\frac{2}{3nh_{n}^{2}}\sum_{k=1}^{n}{\left({\bar{X}}_{kh_{n},n}^{\left(i\right)}-{\bar{X}}_{\left(k-1\right)h_{n},n}^{\left(i\right)}\right)}^{4}}}\\ \; & \;\times \left(\frac{1}{nh_{n}}\sum_{k=1}^{n}{\left({\bar{X}}_{kh_{n},n}^{\left(i\right)}-{\bar{X}}_{\left(k-1\right)h_{n},n}^{\left(i\right)}\right)}^{2}-\frac{1}{nh_{n}}\sum_{2\leq 2k\leq n}{\left({\bar{X}}_{2kh_{n},n}^{\left(i\right)}-{\bar{X}}_{\left(2k-2\right)h_{n},n}^{\left(i\right)}\right)}^{2}\right)\\ \; & =\sqrt{\frac{3/2}{\sum_{k=1}^{n}{\left({\bar{X}}_{kh_{n},n}^{\left(i\right)}-{\bar{X}}_{\left(k-1\right)h_{n},n}^{\left(i\right)}\right)}^{4}}}\\ \; & \;\times \left(\sum_{k=1}^{n}{\left({\bar{X}}_{k,n}^{\left(i\right)}-{\bar{X}}_{k-1,n}^{\left(i\right)}\right)}^{2}-\sum_{2\leq 2k\leq n}{\left({\bar{X}}_{2kh_{n},n}^{\left(i\right)}-{\bar{X}}_{\left(2k-2\right)h_{n},n}^{\left(i\right)}\right)}^{2}\right), \end{array}$$
and we abbreviate $T_{i,n}$ to $T_{n}$ if d=1. Under $H_{0}$, we have the next result.

**Theorem** **2.**
*Under $H_{0}$ and [A1], we have the convergence in law such that*
$$\begin{array}{c} T_{i,n}\to^{L}N\left(0,1\right). \end{array}$$


We also obtain the result to support the consistency of the test.

**Theorem** **3.**
*Under $H_{1}$ and [A1], we have convergence such that for any c∈R,*
$$\begin{array}{cc} \; & P\left(T_{i,n}<c\right)\to 1. \end{array}$$


**Remark** **3.**
*These results are intuitive since the quadratic variation and the reduced one with some appropriate scaling should converge to the same value if $H_{0}$ holds and the quadratic variation with some appropriate scaling should converge to the value which is smaller than the value which the reduced one with scaling converge to.*


Hence when we set the significance level $\alpha_{\mathrm{sig}}\in \left(0,1\right)$, then we have the rejection region
$$\begin{array}{cc} \; & T_{i,n}<\Phi^{-1}\left(\alpha_{\mathrm{sig}}\right) \end{array}$$
where Φ is the distribution function of the standard Gaussian distribution. Theorem 3 supports the consistency of the test.

This test is essential in terms of examining the validity to consider the scheme of convolutional observation: if ρ=0, then the ordinary observation scheme can be applied, but if ρ≠0, then we have the motivation to consider the convolutional observation scheme.

## 4. Least Square Estimation of the Diffusion and Drift Parameters

Let us set the least-square quasi-likelihood functions such that
$$\begin{array}{cc} H_{1,n}\left(\alpha |\rho \right) & :=-\sum_{k=1}^{n}{∥\frac{1}{h_{n}}{\left({\bar{X}}_{kh_{n},n}-{\bar{X}}_{\left(k-1\right)h_{n},n}\right)}^{⊗2}-G\left({\bar{X}}_{\left(k-1\right)h_{n},n},\alpha |\rho \right)∥}^{2},\\ H_{2,n}\left(\beta |\rho \right) & :=-\sum_{k=\left[\max_{i}\rho^{\left(i\right)}\right]+2}^{n}\frac{1}{h_{n}}{\left|{\bar{X}}_{kh_{n},n}-{\bar{X}}_{\left(k-1\right)h_{n},n}-h_{n}b\left({\bar{X}}_{\left(k-2-\left[\max_{i}\rho^{\left(i\right)}\right]\right)h_{n},n},\beta \right)\right|}^{2}, \end{array}$$
and the least-square estimators ${\hat{\alpha }}_{n}$ and ${\hat{\beta }}_{n}$ satisfying
$$\begin{array}{c} H_{1,n}\left({\hat{\alpha }}_{n}|\rho_{⋆}\right)=\underset{\alpha \in \Theta_{1}}{\sup}H_{1,n}\left(\alpha |\rho_{⋆}\right),\;H_{2,n}\left({\hat{\beta }}_{n}|\rho_{⋆}\right)=\underset{\beta \in \Theta_{2}}{\sup}H_{2,n}\left(\beta |\rho_{⋆}\right). \end{array}$$
when $\rho_{⋆}$ is known, and
$$\begin{array}{c} H_{1,n}\left({\hat{\alpha }}_{n}|{\hat{\rho }}_{n}\right)=\underset{\alpha \in \Theta_{1}}{\sup}H_{1,n}\left(\alpha |{\hat{\rho }}_{n}\right),\;H_{2,n}\left({\hat{\beta }}_{n}|{\hat{\rho }}_{n}\right)=\underset{\beta \in \Theta_{2}}{\sup}H_{2,n}\left(\beta |{\hat{\rho }}_{n}\right). \end{array}$$
when $\rho_{⋆}$ is unknown.

**Theorem** **4.**
*Under [A1]–[A3], ${\hat{\alpha }}_{n}$ and ${\hat{\beta }}_{n}$ are consistent, i.e., ${\hat{\alpha }}_{n}\to^{P}\alpha_{⋆}$ and ${\hat{\beta }}_{n}\to^{P}\beta_{⋆}$.*


**Remark** **4.**
*The function G and $f_{G}$ are indeed complex and confusing; hence, we can consider some alternative estimation methods with subsampling or pre-averaging. However, these methods also have the problem what size of subsampling or pre-averaging is proper and the result of the estimation can be dependent on tuning the subsampling size or pre-averaging one. Therefore, our work proposes the estimation method which uses the observation without subsampling or pre-averaging.*


## 5. Simulations

In this simulation section, we only consider the case where ρ is unknown and should be estimated by data with the method proposed in Section 3.

### 5.1. 1-Dimensional Simulation

We examine the following 1-dimensional stochastic differential equation whose solution is a 1-dimensional Ornstein–Uhlenbeck (OU) process:$$\begin{array}{c} dX_{t}=\left(\beta^{\left(1\right)}X_{t}+\beta^{\left(2\right)}\right)dt+\alpha dw_{t},\;X_{-\lambda }=0, \end{array}$$
$\alpha \in \Theta_{1}:=\left[0.01,10\right]$, $\beta \in \Theta_{2}:=\left[-10,-0.01\right]\times \left[-10,10\right]$, and $\lambda =10^{-7/3}$. The procedure of the simulation is as follows: in the first place we iterate an approximated OU process by Euler–Maruyama scheme, for example, see [26] with simulation parameters $n_{\mathrm{sim}}=10^{5+m}$, $h_{\mathrm{sim}}=10^{-10/3-m}$, $T_{\mathrm{sim}}=10^{5/3}$ where m∈N is a parameter to determine the precision of approximation; secondly, we give the approximation of convolution by summation such that
$$\begin{array}{c} {\bar{X}}_{ih_{n},n}≊\left\{\begin{array}{cc} \frac{1}{\left[10^{m}\rho \right]}\sum_{k=0}^{10^{m}-1}X_{ih_{n}-kh_{\mathrm{sim}}} & \;\mathrm{if}\;\left[10^{m}\rho \right]\geq 1,\\ X_{ih_{n}} & \;\mathrm{if}\;\left[10^{m}\rho \right]<0, \end{array}\right. \end{array}$$
where i=0,…,n, the sampling frequency $h_{n}=10^{-10/3}$ and $n=10^{5/3}$. In this Section 5.1, we fix the true value of α and β as $\alpha_{⋆}=3$ and $\beta_{⋆}={\left[-2,1\right]}^{T}$, and change the true value of $\rho \in \Theta_{\rho }:=\left[0,100\right]$ to see the corresponding changes of performance of estimation for ξ, and test for ρ in comparison to estimation by an existent method called local Gaussian approximation (LGA) for parametric estimation of discretely observed diffusion processes, e.g., see [4] which does not concern convolutional observation. All the numbers of iterations for different ρ’s are 1000.

In the first place, we see the estimation and test with small values of $\rho_{⋆}$ such that $\rho_{⋆}=0,0.1,0.2,\ldots ,1$ to observe how the performance of statistics changes by difference in ρ. Table 1 summarises the results of simulation of ${\hat{\rho }}_{n}$ for ρ’s with respective empirical means and root mean square error (RMSE).

We can see the proposed estimator ${\hat{\rho }}_{n}$ works well for small ρ. With respect to the performance of the test statistic $T_{n}$ proposed in Section 3.2, Table 2 shows the empirical ratio of the number of iterations whose $T_{n}$ is lower than some typical critical values where Φ indicates the distribution function of 1-dimensional standard Gaussian distribution as well as the maximum value of $T_{n}$ in 1000 iterations.

Even for ρ=0.1, the simulation result supports the theoretical discussion of the test with consistency. Because $\Phi \left(10^{-16}\right)=-8.222$, all the iterations with ρ≥0.3 result in rejection of $H_{0}$ with substantially significance level $10^{-16}$. Let us see the estimation for α and β by our proposal method and that by the LGA in Table 3.

Note that the biases of the estimation by LGA increase as the true value of ρ gets larger, while the estimation by our proposal method is not influenced by the true value of ρ. This result of the simulation supports the theoretical discussion in Section 4 stating the consistency of ${\hat{\theta }}_{n}$, and necessity to consider the convolutional observation scheme where the LGA method does not work properly.

Secondly, we consider the estimation and test with large $\rho_{⋆}$ such that $\rho_{⋆}$ = 10, 15, 20 to see if our proposal methods work even for large ρ. We note that the maximum values of $T_{n}$ for ρ = 10, 15, 20 in 1000 iterations are −55.091, −68.462 and −79.105, and hence we can detect the smoothed observation easily. Table 4 shows the empirical means and RMSEs of ${\hat{\rho }}_{n}$ for ρ = 10, 15, 20 and we can see that the RMSEs increase as ρ’s increase; it indicates the difficulty to estimate ρ accurately when $\rho_{⋆}$ is large.

Table 5 summarises the estimation for θ by means and RMSE, and tells us that although the large RMSE of ${\hat{\rho }}_{n}$ results in increase of RMSE of ${\hat{\alpha }}_{n}$, estimation by our method is substantially better than that by LGA.

### 5.2. 2-Dimensional Simulation

We consider the following 2-dimensional stochastic differential equation whose solution is a 2-dimensional OU process:$$\begin{array}{cc} d\left[\begin{array}{c} X_{t}^{\left(1\right)}\\ X_{t}^{\left(2\right)} \end{array}\right]\; & =\left(\left[\begin{array}{cc} \beta^{\left(1\right)} & \beta^{\left(2\right)}\\ \beta^{\left(4\right)} & \beta^{\left(5\right)} \end{array}\right]\left[\begin{array}{c} X_{t}^{\left(1\right)}\\ X_{t}^{\left(2\right)} \end{array}\right]+\left[\begin{array}{c} \beta^{\left(3\right)}\\ \beta^{\left(6\right)} \end{array}\right]\right)dt+\left[\begin{array}{cc} \alpha^{\left(1\right)} & \alpha^{\left(2\right)}\\ \alpha^{\left(2\right)} & \alpha^{\left(3\right)} \end{array}\right]dw_{t},X_{-\lambda }=\left[\begin{array}{c} 0\\ 0 \end{array}\right], \end{array}$$$\lambda =10^{-7/3}$. The simulation is conducted with the settings as follows: firstly, we iterate the OU process by Euler–Maruyama scheme with the simulation sample size $n_{\mathrm{sim}}=10^{5+m}$, $T_{\mathrm{sim}}=10^{5/3}$ and discretisation step $h_{\mathrm{sim}}=10^{-10/3-m}$, where m=2 is the precision parameter for approximation of convolution; in the second place, we approximate the convoluted process with summation such that
$$\begin{array}{c} {\bar{X}}_{ih_{n},n}^{\left(j\right)}≊\left\{\begin{array}{cc} \frac{1}{\left[10^{m}\rho^{\left(j\right)}\right]}\sum_{k=0}^{10^{m}-1}X_{ih_{n}-kh_{\mathrm{sim}}}^{\left(j\right)} & \;\mathrm{if}\;\left[10^{m}\rho^{\left(j\right)}\right]\geq 1,\\ X_{ih_{n}}^{\left(j\right)} & \;\mathrm{if}\;\left[10^{m}\rho^{\left(j\right)}\right]<0, \end{array}\right. \end{array}$$
where i=0,…,n, j=1,2, the sampling scheme for inference is defined as $n=10^{5}$ and $h_{n}=10^{-10/3}$; the true value of ρ, α and β are set as $\rho_{⋆}={\left[2,4\right]}^{T}$, $\alpha_{⋆}={\left[2,0,3\right]}^{T}$, $\beta_{⋆}={\left[-2,-0.4,0,0.1,-3,5\right]}^{T}$; the parameter spaces are defined as $\Theta_{\rho }={\left[0,10\right]}^{2}$, $\Theta_{1}=\left[1+10^{-8},10\right]\times \left[-1+10^{-8},1-10^{-8}\right]\times \left[1+10^{-8},10\right]$, and $\Theta_{2}={\left[-10,10\right]}^{6}$; the total iteration number is set to 1000.

Table 6 summarises the estimation for ρ with the method proposed in Section 3 (the inverse of *r* is computationally obtained) with empirical means and empirical RMSEs of ${\hat{\rho }}_{n}$ in 1000 iterations. We can see that ${\hat{\rho }}_{n}$ is sufficiently precise to estimate the true value of ρ indeed in this result, which is significant to estimate the other parameters α and β.

We also note that the maximum values of the test statistics for smoothed observation proposed in Section 3.2 in 1000 iterations are −17.947 and −33.159 for each axis. The *p*-value for them are smaller than $10^{-16}$; therefore, we can conclude that it is possible to detect the smoothed observation with the proposed test statistic in the case $\rho^{\left(i\right)}=2.0$ if d=2 from this result.

With respect to the estimation for α and β, we compare the estimates by our proposal method with that by LGA which does not concern convolutional observation. Table 7 is the summary for α estimate by both the methods:

We can see that the estimation precision for α by our proposal outperforms those by LGA. This results support validity of our estimation method if we have convolutional observation for diffusion processes. Regarding β, the simulation result is summarised in Table 8:

Though the estimation for $\beta^{\left(3\right)}$ by our method has the smaller bias in comparison to that by LGA, the RMSE of our method is larger than that of LGA; in the estimation for other parameters, our proposal method outperforms the method by LGA. We can conclude that our proposal for estimation of α and β concerning convolutional observation performs better than that not considering this observation scheme.

## 6. Real Data Analysis

In this section, we analyse the EEG dataset named S02E.mat provided in “2. Two class motor imagery (002-2014)” of BNCI Horizon 2020 [20]. The datasets including S02E.mat are also studied by Steryl et al. [27].

### 6.1. Estimation and Test for the Smoothing Parameters

In the first place, we pick up the first 15 axes of the dataset and compute ${\hat{\rho }}_{n}$ and $T_{n}$ proposed in Section 3.1 and Section 3.2 respectively. The results are shown in Table 9.

We can observe that all the 15 time series data have the smoothing parameter ρ>0 with statistical significance when we assume ordinary significance levels. These results motivate us to use our methods for parametric estimation proposed in Section 4 when we fit stochastic differential equations for these data.

### 6.2. Parametric Estimation for a Diffusion Process

We fit a 1-dimensional OU process for the time series data in the 2nd column of the data file S02E.mat with 512 Hz observation for 222 s (the plot of the path can be seen in Figure 1), whose ${\hat{\rho }}_{n}=1.037$ is the largest among those for the 15 axes and it is larger than 0 with statistical significance. According to the simulation result shown in Section 5.1, this size of the smoothing parameter gives critical biases when we estimate α and β with LGA method not concerning convolutional observation scheme.

The stochastic differential equation with parameters $\alpha \in \Theta_{1}:=\left[0.01,200\right]$ and $\beta \in \Theta_{2}:=\left[-100,-0.01\right]\times \left[-100,100\right]$ is defined as follows:$$\begin{array}{c} dX_{t}=\left(\beta^{\left(1\right)}X_{t}+\beta^{\left(2\right)}\right)dt+\alpha dw_{t},\;X_{-\lambda }=x_{-\lambda }. \end{array}$$

We set 5 s as the time unit: hence n= 113,664 and $h_{n}=1/\left(5\times 512\right)$. If we fit the OU process with the LGA method, i.e., we do not concern convolutional observation scheme, we obtain the fitting result such that
$$\begin{array}{c} dX_{t}=\left(\left(-17.378\right)X_{t}+\left(-1.091\right)\right)dt+\left(122.892\right)dw_{t},\;X_{-\lambda }=x_{-\lambda }. \end{array}$$

In the next place, we fit α and β with the least square method proposed in Section 4, and then we have the next fitting result:$$\begin{array}{c} dX_{t}=\left(\left(-2.146\right)X_{t}+\left(0.552\right)\right)dt+\left(151.919\right)dw_{t},\;X_{-\lambda }=x_{-\lambda }. \end{array}$$

It is worth noting that this fitting result is substantially different to that by LGA as shown above: hence these results indicate the significance to examine if the observation is convoluted with the smoothing parameter ρ>0 and otherwise the estimation is strongly biased.

## 7. Summary

We have discussed the convolutional observation scheme which deals with the smoothness of observation in comparison to ordinary diffusion processes. The first contribution is to propose this new observation scheme with the statistical test to confirm whether this scheme is valid in real data. The second one is to prove consistency of the estimator ${\hat{\rho }}_{n}$ for the smoothing parameter ρ, those for parameters in diffusion and drift coefficient, i.e., α and β, of the latent diffusion process $\left\{X_{t}\right\}$. Thirdly, we have examined the performance of those estimators and the test statistics in computational simulation, and verified these statistics work well in realistic settings. In the fourth place, we have shown a real example of observation where ρ≠0 holds with statistical significance.

If we combine the test for noise detection by Nakakita and Uchida [28] and that for smoothed observation proposed in this paper, we can test if the observed process is diffusion or not in terms that the observation is noisy or smoothed. Note that the realised volatilities of the noisy observation of diffusion processes increase as observation frequency increases while those of the smoothed observation decreases as the frequency grows. On that point, the noisy observation in ordinary meaning and the smoothed one are ‘opposite’ to each other.

These contributions, especially the third one, will cultivate the motivation to study statistical approaches for convolutionally observed diffusion processes furthermore, such as estimation of kernel function *V* appearing in the convoluted diffusion ${\bar{X}}_{t}:=\left(V*X\right)\left(t\right)$, test theory for parameters α and β as likelihood-ratio-type test statistics, for example, see [29,30], large deviation inequalities for quasi-likelihood functions and associated discussion of Bayes-type estimators, e.g., [6,31,32,33]. By these future works, it is expected that the applicability of stochastic differential equations in real data analysis and contributions to the areas with high frequency observation of phenomena such as EEG will be enhanced.

## 8. Proofs

### 8.1. the Results for Some Laws of Large Numbers

In this subsection, we give the notations and statements of propositions without proofs except for Proposition 3: the detailed proofs are given in Appendix A. We assume Δ≤λ, k∈N,M>0, and consider a class of ${\bar{R}}^{k}⊗{\bar{R}}^{d}$-valued kernel functions on R denoted as $K\left(\Delta ,k,M\right)$ such that for all $\Phi_{\Delta }\in K\left(\Delta ,k,M\right)$, it holds:$$\begin{array}{cc} \left(i\right)\; & \mathrm{supp}\Phi_{\Delta }\subset \left[0,\Delta \right],\\ \left(\mathrm{ii}\right)\; & \mathrm{for}\;\mathrm{all}\;f:\left[0,\Delta \right]\times \Omega \to R^{k},\omega \in \Omega ,\left|\int_{0}^{\Delta }\Phi_{\Delta }\left(\Delta -s\right)f\left(s,\omega \right)ds\right|\leq M\underset{s\in \left[0,\Delta \right]}{\sup}\left|f\left(s,\omega \right)\right|\\ \left(\mathrm{iii}\right)\; & \mathrm{for}\;\mathrm{all}\;t_{0}\geq -\lambda ,f:R^{d}\to R\mathrm{which}\;\mathrm{is}\;\mathrm{continuous}\;\mathrm{and}\;\mathrm{at}\;\mathrm{most}\;\mathrm{polynomial}\;\mathrm{growth},\\ \; & E\left[\left.\int_{t-\Delta }^{t}\Phi_{\Delta }\left(t-s\right)f\left(X_{s}\right)ds\right|F_{t_{0}}\right]=\int_{t-\Delta }^{t}\Phi_{\Delta }\left(t-s\right)E\left[\left.f\left(X_{s}\right)\right|F_{t_{0}}\right]ds. \end{array}$$

**Remark** **5.**
*Note one sufficient condition for $\Phi_{\Delta }\in K\left(\Delta ,k,M\right)$ is (i) $\Phi_{\Delta }:R\to R^{k}⊗R^{d}$, (ii) $\mathrm{supp}\Phi_{\Delta }\subset \left[0,\Delta \right]$, (iii) $\int_{0}^{\Delta }∥\Phi_{\Delta }\left(\Delta -s\right)∥ds\leq M$ and (iv) $B\left(\left[0,\Delta \right]\right)$-measurable since*
$$\begin{array}{cc} \left|\int_{0}^{\Delta }\Phi_{\Delta }\left(\Delta -s\right)f\left(s,\omega \right)ds\right| & \leq \int_{0}^{\Delta }∥\Phi_{\Delta }\left(\Delta -s\right)∥\left|f\left(s,\omega \right)\right|ds\leq M\underset{s\in \left[0,\Delta \right]}{\sup}\left|f\left(s,\omega \right)\right| \end{array}$$
*for the Cauchy–Schwarz inequality, and Fubini’s theorem.*


It is easily checked that $V_{\rho ,h_{n}}\in K\left(\max_{i=1,\ldots ,d}\rho_{i}h_{n},d,d\right)$.

Let *p* denote an integer such that $\sup_{n\in N}ph_{n}\leq \lambda$, $\Delta_{n}:=ph_{n}$. We set the sequence of the kernels ${\left\{\Phi_{\Delta_{n},n}\right\}}_{n\in N}$ such that $\Phi_{\Delta_{n},n}\in K\left(\Delta_{n},d,M\right)$ for some M>0, $\int_{0}^{\Delta_{n}}\Phi_{\Delta_{n},n}ds=I_{d}$ and there exist a matrix $B\in R^{d}⊗R^{d}$ such that
$$\begin{array}{c} ∥\int_{0}^{\Delta_{n}+h_{n}}\left(\Phi_{\Delta_{n},n}\left(\left(\Delta_{n}+h_{n}\right)-s\right)-\Phi_{\Delta_{n},n}\left(\Delta_{n}-s\right)\right)sds-h_{n}B∥\leq Ch_{n}^{2}{\left(1+\left|x\right|\right)}^{C}, \end{array}$$
a set $L\subset \left\{0,\ldots ,p\right\}$ such that there exist functions $D_{\ell }:R^{d}\to R^{d}⊗R^{d}$ for ℓ∈L such that
$$\begin{array}{cc} \; & ∥E\left[\left(\int_{0}^{\Delta_{n}+\left(1+\ell \right)h_{n}}\Phi_{\Delta_{n},n}\left(\Delta_{n}-s_{1}\right)\left(\int_{0}^{s_{1}}a\left(x\right)dw_{s_{2}}\right)ds_{1}\right)\right.\\ \; & \;\left(\int_{0}^{\Delta_{n}+\left(1+\ell \right)h_{n}}\left(\Phi_{\Delta_{n},n}\left(\left(\Delta_{n}+\left(1+\ell \right)h_{n}\right)-s_{1}\right)-\Phi_{\Delta_{n},n}\left(\Delta_{n}+\ell h_{n}-s_{1}\right)\right)\right.\\ \; & \;\left.{\left.\times \left(\int_{0}^{s_{1}}a\left(x\right)dw_{s_{2}}\right)ds_{1}\right)}^{T}\right]-h_{n}D_{\ell }\left(x\right)∥\\ \; & \leq Ch_{n}^{2}{\left(1+\left|x\right|\right)}^{C}, \end{array}$$
a function $G:R^{d}\to R^{d}⊗R^{d}$ such that
$$\begin{array}{cc} \; & ∥E\left[\left(\int_{0}^{\Delta_{n}+h_{n}}\left(\Phi_{\Delta_{n},n}\left(\left(\Delta_{n}+h_{n}\right)-s_{1}\right)-\Phi_{\Delta_{n},n}\left(\Delta_{n}-s_{1}\right)\right)\left(\int_{0}^{s_{1}}a\left(x\right)dw_{s_{2}}\right)ds_{1}\right)\right.\\ \; & \;\left.{\left(\int_{0}^{\Delta_{n}+h_{n}}\left(\Phi_{\Delta_{n},n}\left(\left(\Delta_{n}+h_{n}\right)-s_{1}\right)-\Phi_{\Delta_{n},n}\left(\Delta_{n}-s_{1}\right)\right)\left(\int_{0}^{s_{1}}a\left(x\right)dw_{s_{2}}\right)ds_{1}\right)}^{T}\right]\\ \; & \;-h_{n}G\left(x\right)∥\\ \; & \leq Ch_{n}^{2}{\left(1+\left|x\right|\right)}^{C}. \end{array}$$

We define
$$\begin{array}{c} {\bar{X}}_{t,n}=\int_{t-\Delta_{n}}^{t}\Phi_{\Delta_{n},n}\left(t-s\right)X_{s}ds, \end{array}$$
and the following random quantities such that
$$\begin{array}{cc} {\bar{\nu }}_{n}\left(f\left(\cdot ,\xi \right)\right) & :=\frac{1}{n}\sum_{i=1}^{n}f\left({\bar{X}}_{ih_{n},n},\xi \right),\\ {\bar{I}}_{\ell ,n}\left(v\left(\cdot ,\xi \right)\right) & :=\frac{1}{nh_{n}}\sum_{i=1+\ell }^{n}v\left({\bar{X}}_{\left(i-1-\ell \right)h_{n},n},\xi \right)\left[{\bar{X}}_{ih_{n},n}-{\bar{X}}_{\left(i-1\right)h_{n},n}-\left(h_{n}B\right)b\left({\bar{X}}_{\left(i-1-\ell \right)h_{n},n}\right)\right],\\ {\bar{Q}}_{n}\left(M\left(\cdot ,\xi \right)\right) & :=\frac{1}{nh_{n}}\sum_{i=1}^{n}M\left({\bar{X}}_{\left(i-1\right)h_{n},n},\xi \right)\left[{\left({\bar{X}}_{ih_{n},n}-{\bar{X}}_{\left(i-1\right)h_{n},n}\right)}^{⊗2}\right], \end{array}$$
where $f:R^{d}\times \Xi \to R$, $v:R^{d}\times \Xi \to R^{d}$, $M:R^{d}\times \Xi \to R^{d}⊗R^{d}$ are in $C^{2}$-class, and their first and second derivatives and themselves are at most polynomial growth uniformly in ξ∈Ξ.

**Proposition** **1.**
*Under [A1], ${\bar{\nu }}_{n}\left(f\left(\cdot ,\xi \right)\right)\to^{P}\nu_{0}\left(f\left(\cdot ,\xi \right)\right)$ uniformly in ξ∈Ξ.*


**Proposition** **2.**
*If ℓ∈L and [A1] hold, ${\bar{I}}_{\ell ,n}\left(v\left(\cdot ,\xi \right)\right)\to^{P}\nu_{0}\left(\partial_{x}v\left[D_{\ell }^{T}\right]\left(\cdot ,\xi \right)\right)$ uniformly in ξ∈Ξ.*


**Proposition** **3.**
*Under [A1], ${\bar{Q}}_{n}\left(M\left(\cdot ,\xi \right)\right)\to^{P}\nu_{0}\left(M\left[G\right]\left(\cdot ,\xi \right)\right)$ uniformly in ξ∈Ξ.*


We set $p=\left[\bar{\rho }\right]+1$, $\Delta_{n}=ph_{n}$ and see the evaluation of *B*, $D_{\ell }$ and *G* when setting our kernel $\left\{\Phi_{\Delta ,n}\right\}=\left\{V_{\rho ,h_{n}}\right\}$ as follows (for the derivation of the evaluation, see the Appendix A): we have $\Delta_{n}=ph_{n}$, $B=I_{d}$, $D_{0}\left(x\right)={\left.D_{0}\left(x|\rho \right)\right|}_{\rho =\rho_{⋆}}$, where $D_{0}^{\left(i,j\right)}\left(x|\rho \right)=A^{\left(i,j\right)}\left(x\right)f_{D_{0}}\left(\rho^{\left(i\right)},\rho^{\left(j\right)}\right)$,
$$\begin{array}{cc} f_{D_{0}}\left(\rho^{\left(i\right)},\rho^{\left(j\right)}\right)\; & :=\left\{\begin{array}{cc} 0 & \;\mathrm{if}\;\rho^{\left(j\right)}=0,\\ \frac{\rho^{\left(j\right)}}{2} & \;\mathrm{if}\;\rho^{\left(i\right)}=0,\rho^{\left(j\right)}\in \left(0,1\right],\\ \frac{2\rho^{\left(j\right)}-1}{2\rho^{\left(j\right)}} & \;\mathrm{if}\;\rho^{\left(i\right)}=0,\rho^{\left(j\right)}\in \left(1,\bar{\rho }\right],\\ \frac{6\rho^{\left(i\right)}\rho^{\left(j\right)}-3{\left(\rho^{\left(i\right)}\right)}^{2}-3\rho^{\left(i\right)}}{6\rho^{\left(i\right)}\rho^{\left(j\right)}} & \;\mathrm{if}\;\rho^{\left(i\right)}>0,\rho^{\left(i\right)}+1<\rho^{\left(j\right)},\\ \frac{{\left(\rho^{\left(i\right)}-\rho^{\left(j\right)}\right)}^{3}+3{\left(\rho^{\left(j\right)}\right)}^{2}-3\rho^{\left(j\right)}+1}{6\rho^{\left(i\right)}\rho^{\left(j\right)}} & \;\mathrm{if}\;\rho^{\left(j\right)}>1,\rho^{\left(i\right)}<\rho^{\left(j\right)}\leq \rho^{\left(i\right)}+1,\\ \frac{3{\left(\rho^{\left(j\right)}\right)}^{2}-3\rho^{\left(j\right)}+1}{6\rho^{\left(i\right)}\rho^{\left(j\right)}} & \;\mathrm{if}\;\rho^{\left(j\right)}>1,\rho^{\left(i\right)}\geq \rho^{\left(j\right)},\\ \frac{{\left(\rho^{\left(i\right)}-\rho^{\left(j\right)}\right)}^{3}+{\left(\rho^{\left(j\right)}\right)}^{3}}{6\rho^{\left(i\right)}\rho^{\left(j\right)}} & \;\mathrm{if}\;\rho^{\left(j\right)}\in \left(0,1\right],\rho^{\left(i\right)}\in \left(0,\rho^{\left(j\right)}\right),\\ \frac{{\left(\rho^{\left(j\right)}\right)}^{3}}{6\rho^{\left(i\right)}\rho^{\left(j\right)}} & \;\mathrm{if}\;\rho^{\left(j\right)}\in \left(0,1\right],\rho^{\left(i\right)}\geq \rho^{\left(j\right)}, \end{array}\right. \end{array}$$$D_{\ell }=O$ for $\ell \geq \left[\max_{i=1,\ldots ,d}\rho_{*}^{\left(d\right)}\right]+1$ because of independent increments of the Wiener process, and $G\left(x\right)={\left.G\left(x|\rho \right)\right|}_{\rho =\rho_{⋆}}$ where $G\left(x|\rho \right)={\left.G\left(x,\alpha |\rho \right)\right|}_{\alpha =\alpha_{⋆}}$.

### 8.2. Proof of the Results in Section 3.1

**Proof** **of** **Lemma** **1.**By following the proof of the Proposition 3, it is sufficient to evaluate
$$\begin{array}{cc} \; & \int_{0}^{\left(p+2\right)h_{n}}\int_{0}^{\left(p+2\right)h_{n}}A^{\left(i,i\right)}\left(x\right)\min\left\{s,s^{\prime }\right\}\left(V_{\rho ,h_{n}}^{\left(i,i\right)}\left(\left(p+2\right)h_{n}-s\right)-V_{\rho ,h_{n}}^{\left(i,i\right)}\left(ph_{n}-s\right)\right)\\ \; & \;\left(V_{\rho ,h_{n}}^{\left(i,i\right)}\left(\left(p+2\right)h_{n}-s^{\prime }\right)-V_{\rho ,h_{n}}^{\left(i,i\right)}\left(ph_{n}-s^{\prime }\right)\right)ds^{\prime }ds \end{array}$$
for the asymptotic behaviour of the reduced quadratic variation by choosing
$$\begin{array}{c} M^{\left(\ell_{1},\ell_{2}\right)}=\left\{\begin{array}{cc} 1 & \mathrm{if}\;\ell_{1}=\ell_{2}=i,\\ 0 & \mathrm{otherwise}. \end{array}\right. \end{array}$$If $\rho^{\left(i\right)}=0$,
$$\begin{array}{cc} \; & \int_{0}^{\left(p+2\right)h_{n}}\int_{0}^{\left(p+2\right)h_{n}}A^{\left(i,i\right)}\left(x\right)\min\left\{s,s^{\prime }\right\}\left(V_{\rho ,h_{n}}^{\left(i,i\right)}\left(\left(p+2\right)h_{n}-s\right)-V_{\rho ,h_{n}}^{\left(i,i\right)}\left(ph_{n}-s\right)\right)\\ \; & \;\left(V_{\rho ,h_{n}}^{\left(i,i\right)}\left(\left(p+2\right)h_{n}-s^{\prime }\right)-V_{\rho ,h_{n}}^{\left(i,i\right)}\left(ph_{n}-s^{\prime }\right)\right)ds^{\prime }ds\\ \; & =A^{\left(i,i\right)}\left(x\right)\int_{0}^{\left(p+2\right)h_{n}}\int_{0}^{\left(p+2\right)h_{n}}\min\left\{s,s^{\prime }\right\}\left(\delta \left(\left(p+2\right)h_{n}-s\right)-\delta \left(ph_{n}-s\right)\right)\\ \; & \;\left(\delta \left(\left(p+2\right)h_{n}-s^{\prime }\right)-\delta \left(ph_{n}-s^{\prime }\right)\right)ds^{\prime }ds\\ \; & =A^{\left(i,i\right)}\left(x\right)\left(\left(p+2\right)h_{n}-2ph_{n}+ph_{n}\right)\\ \; & =2h_{n}A^{\left(i,i\right)}\left(x\right), \end{array}$$
and if $\rho^{\left(i\right)}\in \left(0,\bar{\rho }\right]$,
$$\begin{array}{cc} \; & \int_{0}^{\left(p+2\right)h_{n}}\int_{0}^{\left(p+2\right)h_{n}}A^{\left(i,i\right)}\left(x\right)\min\left\{s,s^{\prime }\right\}\left(V_{\rho ,h_{n}}^{\left(i,i\right)}\left(\left(p+2\right)h_{n}-s\right)-V_{\rho ,h_{n}}^{\left(i,i\right)}\left(ph_{n}-s\right)\right)\\ \; & \;\left(V_{\rho ,h_{n}}^{\left(i,i\right)}\left(\left(p+2\right)h_{n}-s^{\prime }\right)-V_{\rho ,h_{n}}^{\left(i,i\right)}\left(ph_{n}-s^{\prime }\right)\right)ds^{\prime }ds\\ \; & =\frac{A^{\left(i,i\right)}\left(x\right)}{{\left(\rho^{\left(i\right)}h_{n}\right)}^{2}}\int_{0}^{\left(p+2\right)h_{n}}\int_{0}^{\left(p+2\right)h_{n}}\min\left\{s,s^{\prime }\right\}\\ \; & \;\times \left(1_{\left[0,\rho^{\left(i\right)}h_{n}\right]}\left(\left(p+2\right)h_{n}-s\right)-1_{\left[0,\rho^{\left(i\right)}h_{n}\right]}\left(ph_{n}-s\right)\right)\\ \; & \;\times \left(1_{\left[0,\rho^{\left(i\right)}h_{n}\right]}\left(\left(p+2\right)h_{n}-s^{\prime }\right)-1_{\left[0,\rho^{\left(i\right)}h_{n}\right]}\left(ph_{n}-s^{\prime }\right)\right)ds^{\prime }ds\\ \; & =\frac{A^{\left(i,i\right)}\left(x\right)}{{\left(\rho^{\left(i\right)}h_{n}\right)}^{2}}\int_{\left(p+2-\rho^{\left(i\right)}\right)h_{n}}^{\left(p+2\right)h_{n}}\int_{\left(p+2-\rho^{\left(i\right)}\right)h_{n}}^{\left(p+2\right)h_{n}}\min\left\{s,s^{\prime }\right\}ds^{\prime }ds\\ \; & \;-\frac{2A^{\left(i,i\right)}\left(x\right)}{{\left(\rho^{\left(i\right)}h_{n}\right)}^{2}}\int_{\left(p+2-\rho^{\left(i\right)}\right)h_{n}}^{\left(p+2\right)h_{n}}\int_{\left(p-\rho^{\left(i\right)}\right)h_{n}}^{ph_{n}}\min\left\{s,s^{\prime }\right\}ds^{\prime }ds\\ \; & \;+\frac{A^{\left(i,i\right)}\left(x\right)}{{\left(\rho^{\left(i\right)}h_{n}\right)}^{2}}\int_{\left(p-\rho^{\left(i\right)}\right)h_{n}}^{ph_{n}}\int_{\left(p-\rho^{\left(i\right)}\right)h_{n}}^{ph_{n}}\min\left\{s,s^{\prime }\right\}ds^{\prime }ds\\ \; & =\frac{A^{\left(i,i\right)}\left(x\right)}{{\left(\rho^{\left(i\right)}h_{n}\right)}^{2}}\int_{\left(p+2-\rho^{\left(i\right)}\right)h_{n}}^{\left(p+2\right)h_{n}}\left(\int_{s}^{\left(p+2\right)h_{n}}sds^{\prime }+\int_{\left(p+2-\rho^{\left(i\right)}\right)h_{n}}^{s}s^{\prime }ds^{\prime }\right)ds\\ \; & \;-\frac{2A^{\left(i,i\right)}\left(x\right)}{{\left(\rho^{\left(i\right)}h_{n}\right)}^{2}}1_{\left(2,\bar{\rho }\right]}\left(\rho^{\left(i\right)}\right)\int_{ph_{n}}^{\left(p+2\right)h_{n}}\int_{\left(p-\rho^{\left(i\right)}\right)h_{n}}^{ph_{n}}s^{\prime }ds^{\prime }ds\\ \; & \;-\frac{2A^{\left(i,i\right)}\left(x\right)}{{\left(\rho^{\left(i\right)}h_{n}\right)}^{2}}1_{\left(2,\bar{\rho }\right]}\left(\rho^{\left(i\right)}\right)\int_{\left(p+2-\rho^{\left(i\right)}\right)h_{n}}^{ph_{n}}\left(\int_{s}^{ph_{n}}sds^{\prime }+\int_{\left(p-\rho^{\left(i\right)}\right)h_{n}}^{s}s^{\prime }ds^{\prime }\right)ds\\ \; & \;-\frac{2A^{\left(i,i\right)}\left(x\right)}{{\left(\rho^{\left(i\right)}h_{n}\right)}^{2}}1_{\left(0,2\right]}\left(\rho^{\left(i\right)}\right)\int_{\left(p+2-\rho^{\left(i\right)}\right)h_{n}}^{\left(p+2\right)h_{n}}\int_{\left(p-\rho^{\left(i\right)}\right)h_{n}}^{ph_{n}}s^{\prime }ds^{\prime }ds\\ \; & \;+\frac{A^{\left(i,i\right)}\left(x\right)}{{\left(\rho^{\left(i\right)}h_{n}\right)}^{2}}\int_{\left(p-\rho^{\left(i\right)}\right)h_{n}}^{ph_{n}}\left(\int_{s}^{ph_{n}}sds^{\prime }+\int_{\left(p-\rho^{\left(i\right)}\right)h_{n}}^{s}s^{\prime }ds^{\prime }\right)ds\\ \; & =\left\{\begin{array}{cc} 2h_{n}A^{\left(i,i\right)}\left(x\right)\left(1-\frac{\rho^{\left(i\right)}}{6}\right) & \;\mathrm{if}\;\rho^{\left(i\right)}\in \left(0,2\right],\\ 2h_{n}A^{\left(i,i\right)}\left(x\right)\left(\frac{2}{\rho^{\left(i\right)}}-\frac{4}{3{\left(\rho^{\left(i\right)}\right)}^{2}}\right) & \;\mathrm{if}\;\rho^{\left(i\right)}\in \left(2,\bar{\rho }\right]. \end{array}\right. \end{array}$$Hence, we obtain the proof (for details, see the Appendix A). □

**Proof** **of** **Lemma** **2.**Continuity is obvious, and monotonicity is obtained as follows: if $\rho^{\left(i\right)}\in \left(0,1\right]$,
$$\begin{array}{cc} \frac{d}{d\rho^{\left(i\right)}}\left(6-2\rho^{\left(i\right)}\right){\left(6-\rho^{\left(i\right)}\right)}^{-1} & =\left(\left(-2\right)\left(6-\rho^{\left(i\right)}\right)-\left(6-2\rho^{\left(i\right)}\right)\left(-1\right)\right){\left(6-\rho^{\left(i\right)}\right)}^{-2}\\ \; & =\left(-12\right){\left(6-\rho^{\left(i\right)}\right)}^{-2}<0, \end{array}$$
and if $\rho^{\left(i\right)}\in \left(1,2\right]$,
$$\begin{array}{cc} \; & \frac{d}{d\rho^{\left(i\right)}}\left(6\rho^{\left(i\right)}-2\right){\left(6{\left(\rho^{\left(i\right)}\right)}^{2}-{\left(\rho^{\left(i\right)}\right)}^{3}\right)}^{-1}\\ \; & =\left(6\left(6{\left(\rho^{\left(i\right)}\right)}^{2}-{\left(\rho^{\left(i\right)}\right)}^{3}\right)-\left(6\rho^{\left(i\right)}-2\right)\left(12\rho^{\left(i\right)}-3{\left(\rho^{\left(i\right)}\right)}^{2}\right)\right){\left(6{\left(\rho^{\left(i\right)}\right)}^{2}-{\left(\rho^{\left(i\right)}\right)}^{3}\right)}^{-2}\\ \; & =6\rho^{\left(i\right)}\left(-7\rho^{\left(i\right)}+4+2{\left(\rho^{\left(i\right)}\right)}^{2}\right){\left(6{\left(\rho^{\left(i\right)}\right)}^{2}-{\left(\rho^{\left(i\right)}\right)}^{3}\right)}^{-2}<0, \end{array}$$
and if $\rho^{\left(i\right)}\in \left(2,\bar{\rho }\right]$,
$$\begin{array}{cc} \frac{d}{d\rho^{\left(i\right)}}\left(3\rho^{\left(i\right)}-1\right){\left(6\rho^{\left(i\right)}-4\right)}^{-1} & =\left(-18\right){\left(6\rho^{\left(i\right)}-4\right)}^{-2}<0. \end{array}$$The inverse can be obtained directly. □

**Proof** **of** **Theorem** **1.**It follows from Lemma 1, 2 and continuous mapping theorem. □

### 8.3. Proofs of the Results in Section 3.2

**Proof** **of** **Theorem** **2.**We can clearly prove the result by using Lemma 7 in Kessler [4], Proposition 7 in Nakakita and Uchida [28], and Slutsky’s theorem. □

**Proof** **of** **Theorem** **3.**By Lemma 1, there exists a number ℓ<0 such that
$$\begin{array}{c} \frac{1}{nh_{n}}\sum_{k=1}^{n}{\left({\bar{X}}_{kh_{n},n}^{\left(i\right)}-{\bar{X}}_{\left(k-1\right)h_{n},n}^{\left(i\right)}\right)}^{2}-\frac{1}{nh_{n}}\sum_{2\leq 2k\leq n}{\left({\bar{X}}_{2kh_{n},n}^{\left(i\right)}-{\bar{X}}_{\left(2k-2\right)h_{n},n}^{\left(i\right)}\right)}^{2}\to^{P}\ell <0, \end{array}$$
and hence it is sufficient to show that
$$\begin{array}{c} \underset{n\in N}{\sup}E\left[\frac{2}{3nh_{n}^{2}}\sum_{k=1}^{n}{\left({\bar{X}}_{kh_{n},n}^{\left(i\right)}-{\bar{X}}_{\left(k-1\right)h_{n},n}^{\left(i\right)}\right)}^{4}\right]<\infty ; \end{array}$$
and it is obvious that
$$\begin{array}{c} {\left({\bar{X}}_{kh_{n},n}^{\left(i\right)}-{\bar{X}}_{\left(k-1\right)h_{n},n}^{\left(i\right)}\right)}^{4}\leq C{\left({\bar{X}}_{kh_{n},n}^{\left(i\right)}-X_{\left(k-\bar{\rho }-1\right)h_{n}}\right)}^{4}+C{\left({\bar{X}}_{\left(k-1\right)h_{n},n}^{\left(i\right)}-X_{\left(k-\bar{\rho }-1\right)h_{n}}\right)}^{4} \end{array}$$
and
$$\begin{array}{cc} \; & E\left[\left.{\left({\bar{X}}_{kh_{n},n}^{\left(i\right)}-X_{\left(k-\bar{\rho }-1\right)h_{n}}\right)}^{4}\right|F_{\left(k-\bar{\rho }-1\right)h_{n}}\right]\\ \; & =E\left[\left.{\left(\frac{1}{\rho_{⋆}^{\left(i\right)}h_{n}}\int_{\left(k-\rho_{⋆}^{\left(i\right)}\right)h_{n}}^{kh_{n}}\left(X_{s}^{\left(i\right)}-X_{\left(k-\bar{\rho }-1\right)h_{n}}\right)ds\right)}^{4}\right|F_{\left(k-\bar{\rho }-1\right)h_{n}}\right]\\ \; & \leq E\left[\left.{\left(\frac{1}{\rho_{⋆}^{\left(i\right)}h_{n}}\int_{\left(k-\rho_{⋆}^{\left(i\right)}\right)h_{n}}^{kh_{n}}\left|X_{s}^{\left(i\right)}-X_{\left(k-\bar{\rho }-1\right)h_{n}}\right|ds\right)}^{4}\right|F_{\left(k-\bar{\rho }-1\right)h_{n}}\right]\\ \; & \leq E\left[\left.{\left(\frac{1}{\rho_{⋆}^{\left(i\right)}h_{n}}\int_{\left(k-\rho_{⋆}^{\left(i\right)}\right)h_{n}}^{kh_{n}}\underset{s^{\prime }\in \left[\left(k-\rho_{⋆}^{\left(i\right)}\right)h_{n},kh_{n}\right]}{\sup}\left|X_{s^{\prime }}^{\left(i\right)}-X_{\left(k-\bar{\rho }-1\right)h_{n}}\right|ds\right)}^{4}\right|F_{\left(k-\bar{\rho }-1\right)h_{n}}\right]\\ \; & =E\left[\left.\underset{s\in \left[\left(k-\rho_{⋆}^{\left(i\right)}\right)h_{n},kh_{n}\right]}{\sup}{\left|X_{s}^{\left(i\right)}-X_{\left(k-\bar{\rho }-1\right)h_{n}}\right|}^{4}\right|F_{\left(k-\bar{\rho }-1\right)h_{n}}\right]\\ \; & \leq Ch_{n}^{2}{\left(1+\left|X_{\left(k-\bar{\rho }-1\right)h_{n}}\right|\right)}^{C} \end{array}$$
by Proposition A in Gloter [9], and a parallel result holds for ${\left({\bar{X}}_{\left(k-1\right)h_{n},n}^{\left(i\right)}-X_{\left(k-\bar{\rho }-1\right)h_{n}}\right)}^{4}$. Hence we obtain the result. □

### 8.4. Proof of the Results in Section 4

**Proof** **of** **Theorem** **4.**We only deal with the case where $\rho_{⋆}$ is unknown because the discussion for the case where $\rho_{⋆}$ is known is parallel. First of all, we prove the consistency of ${\hat{\alpha }}_{n}$. We obtain that
$$\begin{array}{cc} \; & \left|\frac{1}{n}H_{1,n}\left(\alpha |{\hat{\rho }}_{n}\right)-\frac{1}{n}H_{1,n}\left(\alpha |\rho_{⋆}\right)\right|\\ \; & =\left|-\frac{1}{n}\sum_{k=1}^{n}{∥\frac{1}{h_{n}}{\left({\bar{X}}_{kh_{n},n}-{\bar{X}}_{\left(k-1\right)h_{n},n}\right)}^{⊗2}-G\left({\bar{X}}_{\left(k-1\right)h_{n},n},\alpha |{\hat{\rho }}_{n}\right)∥}^{2}\right.\\ \; & \;\left.+\frac{1}{n}\sum_{k=1}^{n}{∥\frac{1}{h_{n}}{\left({\bar{X}}_{kh_{n},n}-{\bar{X}}_{\left(k-1\right)h_{n},n}\right)}^{⊗2}-G\left({\bar{X}}_{\left(k-1\right)h_{n},n},\alpha |\rho_{⋆}\right)∥}^{2}\right|\\ \; & \leq \left|\frac{2}{nh_{n}}\sum_{k=1}^{n}\left(G\left({\bar{X}}_{\left(k-1\right)h_{n},n},\alpha |{\hat{\rho }}_{n}\right)-G\left({\bar{X}}_{\left(k-1\right)h_{n},n},\alpha |\rho_{⋆}\right)\right)\left[{\left({\bar{X}}_{kh_{n},n}-{\bar{X}}_{\left(k-1\right)h_{n},n}\right)}^{⊗2}\right]\right|\\ \; & \;+\left|\frac{1}{n}\sum_{k=1}^{n}\left({∥G\left({\bar{X}}_{\left(k-1\right)h_{n},n},\alpha |{\hat{\rho }}_{n}\right)∥}^{2}-{∥G\left({\bar{X}}_{\left(k-1\right)h_{n},n},\alpha |\rho_{⋆}\right)∥}^{2}\right)\right|\\ \; & \leq \frac{2}{nh_{n}}\sum_{k=1}^{n}{\left|{\bar{X}}_{kh_{n},n}-{\bar{X}}_{\left(k-1\right)h_{n},n}\right|}^{2}\\ \; & \;\times \sum_{i=1}^{d}\sum_{j=1}^{d}\left|A^{\left(i,j\right)}\left({\bar{X}}_{\left(k-1\right)h_{n},n},\alpha \right)\right|\left|f_{G}\left({\hat{\rho }}_{n}^{\left(i\right)},{\hat{\rho }}_{n}^{\left(j\right)}\right)-f_{G}\left(\rho_{⋆}^{\left(i\right)},\rho_{⋆}^{\left(j\right)}\right)\right|\\ \; & \;+\frac{1}{n}\sum_{k=1}^{n}\sum_{i=1}^{d}\sum_{j=1}^{d}{\left|A^{\left(i,j\right)}\left({\bar{X}}_{\left(k-1\right)h_{n},n},\alpha \right)\right|}^{2}\left|f_{G}^{2}\left({\hat{\rho }}_{n}^{\left(i\right)},{\hat{\rho }}_{n}^{\left(j\right)}\right)-f_{G}^{2}\left(\rho_{⋆}^{\left(i\right)},\rho_{⋆}^{\left(j\right)}\right)\right|\\ \; & \leq \frac{C}{nh_{n}}\sum_{k=1}^{n}{\left(1+\left|{\bar{X}}_{\left(k-1\right)h_{n},n}\right|\right)}^{C}{\left|{\bar{X}}_{kh_{n},n}-{\bar{X}}_{\left(k-1\right)h_{n},n}\right|}^{2}\\ \; & \;\times \sum_{i=1}^{d}\sum_{j=1}^{d}\left|f_{G}\left({\hat{\rho }}_{n}^{\left(i\right)},{\hat{\rho }}_{n}^{\left(j\right)}\right)-f_{G}\left(\rho_{⋆}^{\left(i\right)},\rho_{⋆}^{\left(j\right)}\right)\right|\\ \; & \;+\frac{C}{n}\sum_{k=1}^{n}{\left(1+\left|{\bar{X}}_{\left(k-1\right)h_{n},n}\right|\right)}^{C}\sum_{i=1}^{d}\sum_{j=1}^{d}\left|f_{G}^{2}\left({\hat{\rho }}_{n}^{\left(i\right)},{\hat{\rho }}_{n}^{\left(j\right)}\right)-f_{G}^{2}\left(\rho_{⋆}^{\left(i\right)},\rho_{⋆}^{\left(j\right)}\right)\right|\\ \; & \to^{P}0\;\mathrm{uniformly}\;\mathrm{in}\;\alpha , \end{array}$$
because continuous mapping theorem holds. Therefore, it follows from Proposition 1 and Proposition 3 that
$$\begin{array}{cc} \; & \frac{1}{n}H_{1,n}\left(\alpha |{\hat{\rho }}_{n}\right)-\frac{1}{n}H_{1,n}\left(\alpha_{⋆}|\rho_{⋆}\right)\\ \; & =\frac{2}{nh_{n}}\sum_{k=1}^{n}G\left({\bar{X}}_{\left(k-1\right)h_{n},n},\alpha |\rho_{⋆}\right)\left[{\left({\bar{X}}_{kh_{n},n}-{\bar{X}}_{\left(k-1\right)h_{n},n}\right)}^{⊗2}\right]\\ \; & \;-\frac{1}{n}\sum_{k=1}^{n}{∥G\left({\bar{X}}_{\left(k-1\right)h_{n},n},\alpha |\rho_{⋆}\right)∥}^{2}\\ \; & \;-\frac{2}{nh_{n}}\sum_{k=1}^{n}G\left({\bar{X}}_{\left(k-1\right)h_{n},n},\alpha_{⋆}|\rho_{⋆}\right)\left[{\left({\bar{X}}_{kh_{n},n}-{\bar{X}}_{\left(k-1\right)h_{n},n}\right)}^{⊗2}\right]\\ \; & \;+\frac{1}{n}\sum_{k=1}^{n}{∥G\left({\bar{X}}_{\left(k-1\right)h_{n},n},\alpha_{⋆}|\rho_{⋆}\right)∥}^{2}\\ \; & \;+o_{P}^{*}\left(1\right)\\ \; & \to^{P}V_{1}\left(\alpha |\xi_{⋆}\right)\;\mathrm{uniformly}\;\mathrm{in}\;\alpha \end{array}$$
where $o_{P}^{*}\left(1\right)$ indicates the term converging in probability to zero uniformly in θ. Then we obtain that ${\hat{\alpha }}_{n}\to \alpha_{⋆}$ in the same way as Kessler [4] with Assumption [A3].In the next place, we consider the consistency of ${\hat{\beta }}_{n}$. Firstly, we consider the case $\max_{i}\rho_{⋆}^{\left(i\right)}\in \left(\ell -1,\ell \right)$ for an integer $\ell \in \left\{1,\ldots ,\left[\bar{\rho }\right]+1\right\}$. Then it is sufficient to show
$$\begin{array}{cc} \; & \frac{1}{nh_{n}}H_{2,n}\left(\beta |{\hat{\rho }}_{n}\right)-\frac{1}{nh_{n}}H_{2,n}\left(\beta_{⋆}|\rho_{⋆}\right)\to^{P}V_{2}\left(\beta |\xi_{⋆}\right)\;\mathrm{uniformly}\;\mathrm{in}\;\beta \end{array}$$
due to Assumption [A3]. Because the evaluation $D_{j}\left(x\right)=O$ where $j\geq \left[\max_{i=1,\ldots ,n}\rho_{*}^{\left(i\right)}\right]+1$ using independent increments of the Wiener process, Proposition 1 and Proposition 2 verify
$$\begin{array}{cc} \; & F_{\ell }\left(\beta \right)-\frac{1}{nh_{n}}H_{2,n}\left(\beta_{⋆}|\rho_{⋆}\right)\to^{P}V_{2}\left(\beta |\xi_{⋆}\right)\;\mathrm{uniformly}\;\mathrm{in}\;\beta , \end{array}$$
where
$$\begin{array}{c} F_{j}\left(\beta \right):=-\frac{1}{nh_{n}^{2}}\sum_{k=1+j}^{n}{\left|{\bar{X}}_{kh_{n},n}-{\bar{X}}_{\left(k-1\right)h_{n},n}-h_{n}b\left({\bar{X}}_{\left(k-1-j\right)h_{n},n},\beta \right)\right|}^{2}. \end{array}$$In addition, the exact convergences such that
$$\begin{array}{c} P\left(1_{\left\{\ell \right\}}\left(\left[\max_{i}{\hat{\rho }}_{n}^{\left(i\right)}\right]+1\right)=1\right)\to 1,\;P\left(1_{\left\{j\right\}}\left(\left[\max_{i}{\hat{\rho }}_{n}^{\left(i\right)}\right]+1\right)=1\right)\to 0 \end{array}$$
hold for all j≠ℓ, since for all $j=1,\ldots ,\left[\bar{\rho }\right]+1$,
$$\begin{array}{c} P\left(1_{\left\{j\right\}}\left(\left[\max_{i}{\hat{\rho }}_{n}^{\left(i\right)}\right]+1\right)=1\right)=P\left(\max_{i}{\hat{\rho }}_{n}^{\left(i\right)}\in \left[j-1,j\right)\right). \end{array}$$Therefore, for any ϵ>0,
$$\begin{array}{cc} \; & P\left(\underset{\beta \in \Theta_{2}}{\sup}\left|\frac{1}{nh_{n}}H_{2,n}\left(\beta |{\hat{\rho }}_{n}\right)-\frac{1}{nh_{n}}H_{2,n}\left(\beta_{⋆}|\rho_{⋆}\right)-V_{2}\left(\beta |\xi_{⋆}\right)\right|>\epsilon \right)\\ \; & \leq \sum_{j\neq \ell }P\left(1_{\left\{j\right\}}\left(\left[\max_{i}{\hat{\rho }}_{n}^{\left(i\right)}\right]+1\right)=1\right)\\ \; & \;+P\left(\left\{1_{\left\{\ell \right\}}\left(\left[\max_{i}{\hat{\rho }}_{n}^{\left(i\right)}\right]+1\right)=1\right\}\right.\\ \; & \;\left.\cap \left\{\underset{\beta \in \Theta_{2}}{\sup}\left|F_{\ell }\left(\beta \right)-\frac{1}{nh_{n}}H_{2,n}\left(\beta_{⋆}|\rho_{⋆}\right)-V_{2}\left(\beta |\xi_{⋆}\right)\right|>\epsilon \right\}\right)\\ \; & \to 0. \end{array}$$For the case $\max_{i}\rho_{⋆}^{\left(i\right)}=\ell$ for an integer $\ell =\left\{0,\ldots ,\left[\bar{\rho }\right]+1\right\}$, we similarly obtain
$$\begin{array}{cc} \; & \frac{1}{nh_{n}}H_{2,n}\left(\beta |{\hat{\rho }}_{n}\right)-\frac{1}{nh_{n}}H_{2,n}\left(\beta_{⋆}|\rho_{⋆}\right)\to^{P}V_{2}\left(\beta |\xi_{⋆}\right)\;\mathrm{uniformly}\;\mathrm{in}\;\beta \end{array}$$
because we have
$$\begin{array}{cc} \; & F_{\ell }\left(\beta \right)-\frac{1}{nh_{n}}H_{2,n}\left(\beta_{⋆}|\rho_{⋆}\right)\to^{P}V_{2}\left(\beta |\xi_{⋆}\right)\;\mathrm{uniformly}\;\mathrm{in}\;\beta ,\\ \; & F_{\ell +1}\left(\beta \right)-\frac{1}{nh_{n}}H_{2,n}\left(\beta_{⋆}|\rho_{⋆}\right)\to^{P}V_{2}\left(\beta |\xi_{⋆}\right)\;\mathrm{uniformly}\;\mathrm{in}\;\beta , \end{array}$$
and
$$\begin{array}{cc} \; & P\left(1_{\left\{\ell \right\}}\left(\left[\max_{i}{\hat{\rho }}_{n}^{\left(i\right)}\right]+1\right)+1_{\left\{\ell +1\right\}}\left(\left[\max_{i}{\hat{\rho }}_{n}^{\left(i\right)}\right]+1\right)=1\right)\to 1,\\ \; & P\left(1_{\left\{j\right\}}\left(\left[\max_{i}{\hat{\rho }}_{n}^{\left(i\right)}\right]+1\right)=0\right)\to 1,\;\mathrm{for}\;\mathrm{all}\;j\neq \ell ,\ell +1, \end{array}$$
and it holds that for any ϵ>0,
$$\begin{array}{cc} \; & P\left(\underset{\beta \in \Theta_{2}}{\sup}\left|\frac{1}{nh_{n}}H_{2,n}\left(\beta |{\hat{\rho }}_{n}\right)-\frac{1}{nh_{n}}H_{2,n}\left(\beta_{⋆}|\rho_{⋆}\right)-V_{2}\left(\beta |\xi_{⋆}\right)\right|>\epsilon \right)\\ \; & \leq \sum_{j\neq \ell ,\ell +1}P\left(1_{\left\{j\right\}}\left(\left[\max_{i}{\hat{\rho }}_{n}^{\left(i\right)}\right]+1\right)=1\right)\\ \; & \;+P\left(\left\{1_{\left\{\ell \right\}}\left(\left[\max_{i}{\hat{\rho }}_{n}^{\left(i\right)}\right]+1\right)=1\right\}\right.\\ \; & \;\left.\cap \left\{\underset{\beta \in \Theta_{2}}{\sup}\left|F_{\ell }\left(\beta \right)-\frac{1}{nh_{n}}H_{2,n}\left(\beta_{⋆}|\rho_{⋆}\right)-V_{2}\left(\beta |\xi_{⋆}\right)\right|>\epsilon \right\}\right)\\ \; & \;+P\left(\left\{1_{\left\{\ell +1\right\}}\left(\left[\max_{i}{\hat{\rho }}_{n}^{\left(i\right)}\right]+1\right)=1\right\}\right.\\ \; & \;\left.\cap \left\{\underset{\beta \in \Theta_{2}}{\sup}\left|F_{\ell +1}\left(\beta \right)-\frac{1}{nh_{n}}H_{2,n}\left(\beta_{⋆}|\rho_{⋆}\right)-V_{2}\left(\beta |\xi_{⋆}\right)\right|>\epsilon \right\}\right)\\ \; & \to 0. \end{array}$$Hence it is shown that ${\hat{\beta }}_{n}\to^{P}\beta_{⋆}$ with Assumption [A3]. □

## Figures and Tables

**Figure 1 entropy-22-01031-f001:**
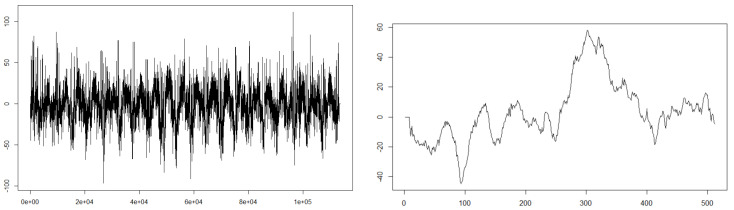
The path of the second column of S02E.mat of BNCI Horizon 2020 [20] for all 222 seconds (**left**) and the first one second (**right**).

**Figure 2 entropy-22-01031-f002:**
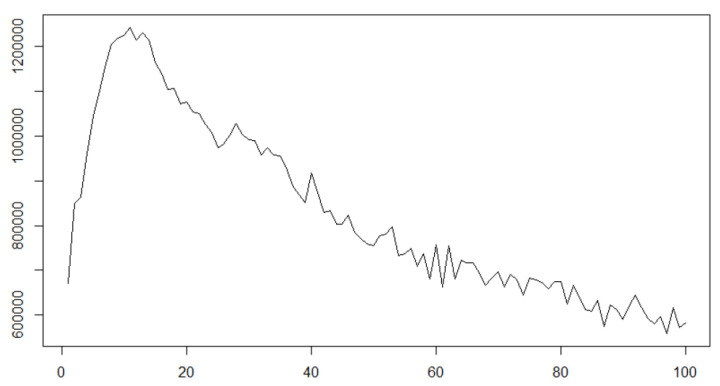
Realised volatilities with subsampling of the 2nd axis of data S02E.mat in two class motor imagery (002-2014) [20].

**Figure 3 entropy-22-01031-f003:**
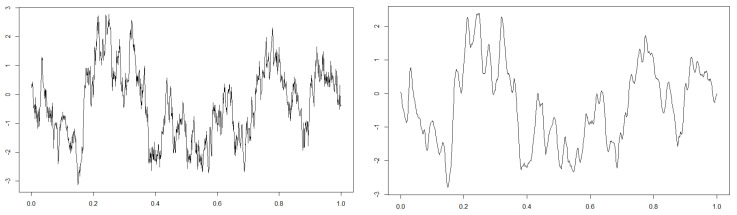
The left figure is the plot of the latent diffusion process, and the right one is that of the convolutionally observed process on $\left[0,1\right]$ respectively.

**Figure 4 entropy-22-01031-f004:**
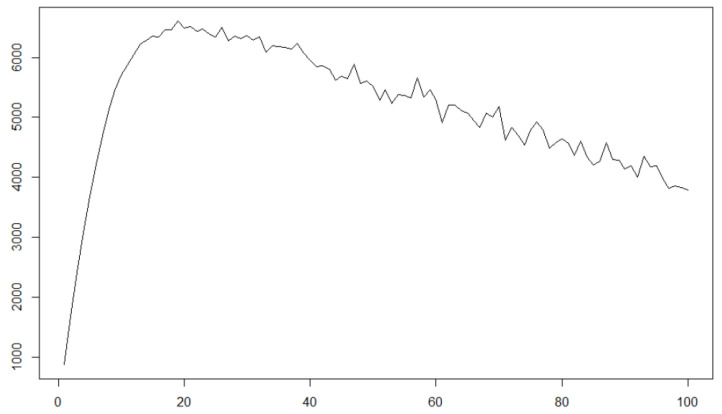
The realised volatilities of the convolutionally observed diffusion process with subsampling.

**Table 1 entropy-22-01031-t001:** Estimation performance of ρ with small ρ.

ρ	0.0	0.1	0.2	0.3	0.4	0.5	0.6	0.7	0.8	0.9	1.0
**mean**	0.00990	0.0971	0.198	0.298	0.398	0.498	0.598	0.699	0.799	0.899	0.999
**RMSE**	(0.0182)	(0.0256)	(0.0235)	(0.0215)	(0.0197)	(0.0180)	(0.0164)	(0.0150)	(0.0135)	(0.0123)	(0.0110)

**Table 2 entropy-22-01031-t002:** Empirical ratio of test statistic $T_{n}$ less than some critical values, and the maximum value of $T_{n}$ in 1000 iterations.

	Empirical Ratio of $T_{n}$ Less Than…					Max. of $T_{n}$
	$\Phi^{-1}\left(0.10\right)$	$\Phi^{-1}\left(0.05\right)$	$\Phi^{-1}\left(0.025\right)$	$\Phi^{-1}\left(0.01\right)$	$\Phi^{-1}\left(0.001\right)$	
ρ=0.0	0.101	0.053	0.025	0.005	0.000	3.060
ρ=0.1	0.989	0.980	0.966	0.914	0.759	−0.710
ρ=0.2	1.000	1.000	1.000	1.000	1.000	−4.593
ρ=0.3	1.000	1.000	1.000	1.000	1.000	−9.341
ρ=0.4	1.000	1.000	1.000	1.000	1.000	−13.985
ρ=0.5	1.000	1.000	1.000	1.000	1.000	−19.152
ρ=0.6	1.000	1.000	1.000	1.000	1.000	−24.816
ρ=0.7	1.000	1.000	1.000	1.000	1.000	−30.848
ρ=0.8	1.000	1.000	1.000	1.000	1.000	−37.557
ρ=0.9	1.000	1.000	1.000	1.000	1.000	−44.829
ρ=1.0	1.000	1.000	1.000	1.000	1.000	−52.759

**Table 3 entropy-22-01031-t003:** Estimation of θ by the proposed method and LGA with small ρ.

		The Proposed Method			LGA		
		α	$\beta^{\left(1\right)}$	$\beta^{\left(2\right)}$	α	$\beta^{\left(1\right)}$	$\beta^{\left(2\right)}$
**True Value**		3.0	−2.0	1.0	3.0	−2.0	1.0
ρ=0.0	mean	3.004	−2.091	1.036	2.999	−2.095	1.037
	RMSE	(0.0109)	(0.318)	(0.497)	(0.00679)	(0.320)	(0.497)
ρ=0.1	mean	2.999	−2.091	1.035	2.949	−2.026	1.003
	RMSE	(0.0146)	(0.319)	(0.496)	(0.0509)	(0.297)	(0.480)
ρ=0.2	mean	2.998	−2.091	1.035	2.898	−1.955	0.967
	RMSE	(0.0142)	(0.319)	(0.496)	(0.102)	(0.290)	(0.464)
ρ=0.3	mean	2.998	−2.092	1.036	2.846	−1.885	0.932
	RMSE	(0.0139)	(0.319)	(0.497)	(0.155)	(0.299)	(0.452)
ρ=0.4	mean	2.998	−2.091	1.036	2.792	−1.815	0.897
	RMSE	(0.0135)	(0.319)	(0.497)	(0.208)	(0.324)	(0.442)
ρ=0.5	mean	2.998	−2.092	1.036	2.738	−1.744	0.862
	RMSE	(0.0132)	(0.319)	(0.497)	(0.262)	(0.361)	(0.436)
ρ=0.6	mean	2.998	−2.091	1.036	2.683	−1.674	0.827
	RMSE	(0.0129)	(0.319)	(0.497)	(0.317)	(0.408)	(0.434)
ρ=0.7	mean	2.998	−2.092	1.036	2.626	−1.604	0.792
	RMSE	(0.0126)	(0.319)	(0.497)	(0.374)	(0.460)	(0.434)
ρ=0.8	mean	2.998	−2.092	1.036	2.568	−1.534	0.757
	RMSE	(0.0124)	(0.319)	(0.496)	(0.432)	(0.517)	(0.439)
ρ=0.9	mean	2.998	−2.092	1.036	2.509	−1.464	0.722
	RMSE	(0.0121)	(0.319)	(0.497)	(0.491)	(0.578)	(0.445)
ρ=1.0	mean	2.998	−2.091	1.036	2.449	−1.394	0.687
	RMSE	(0.0119)	(0.319)	(0.497)	(0.551)	(0.640)	(0.456)

**Table 4 entropy-22-01031-t004:** The performance of ${\hat{\rho }}_{n}$ for ρ=10,15,20 in 1000 iterations.

	ρ=10	ρ=15	ρ=20
mean of ${\hat{\rho }}_{n}$	9.919	14.980	19.751
RMSE of ${\hat{\rho }}_{n}$	(0.145)	(0.240)	(0.409)

**Table 5 entropy-22-01031-t005:** Estimation of θ by the proposed method with large ρ.

		The Proposed Method			LGA		
		α	$\beta^{\left(1\right)}$	$\beta^{\left(2\right)}$	α	$\beta^{\left(1\right)}$	$\beta^{\left(2\right)}$
**True Value**		3.0	−2.0	1.0	3.0	−2.0	1.0
ρ=10	mean	2.989	−2.101	1.030	0.933	−0.204	0.0811
	RMSE	(0.0347)	(0.323)	(0.496)	(2.067)	(1.796)	(0.920)
ρ=15	mean	2.996	−2.095	1.027	0.765	−0.138	0.0473
	RMSE	(0.0475)	(0.321)	(0.495)	(2.235)	(1.862)	(0.953)
ρ=20	mean	2.977	−2.090	1.024	0.664	−0.104	0.0302
	RMSE	(0.0526)	(0.319)	(0.493)	(2.336)	(1.896)	(0.970)

**Table 6 entropy-22-01031-t006:** Summary for ρ estimate.

	$\rho^{\left(1\right)}$	$\rho^{\left(2\right)}$
true value	2.0	4.0
empirical mean	1.988	3.966
empirical RMSE	(0.0207)	(0.0514)

**Table 7 entropy-22-01031-t007:** Summary for α estimate.

		$\alpha^{\left(1\right)}$	$\alpha^{\left(2\right)}$	$\alpha^{\left(3\right)}$
	**True Value**	2.0	0.0	3.0
Our proposal	mean	1.993	0.000256	2.992
	RMSE	(0.0115)	(0.00739)	(0.0213)
LGA	mean	1.295	−0.00320	1.442
	RMSE	(0.705)	(0.0154)	(1.558)

**Table 8 entropy-22-01031-t008:** Summary for β estimate.

		$\beta^{\left(1\right)}$	$\beta^{\left(2\right)}$	$\beta^{\left(3\right)}$	$\beta^{\left(4\right)}$	$\beta^{\left(5\right)}$	$\beta^{\left(6\right)}$
	**True Value**	−2.0	−0.4	0.0	0.1	−3.0	5.0
Our proposal	mean	−2.137	−0.408	−0.0439	0.0788	−3.103	5.091
	RMSE	(0.362)	(0.252)	(0.540)	(0.473)	(0.399)	(0.777)
LGA	mean	−0.917	0.340	−0.326	−0.696	0.221	1.243
	RMSE	(1.093)	(0.802)	(0.386)	(0.804)	(3.242)	(3.765)

**Table 9 entropy-22-01031-t009:** The values of ${\hat{\rho }}_{n}$ and $T_{n}$ for the first 15 axes of S02.mat by BNCI Horizon 2020 [20].

	1st axis	2nd axis	3rd axis	4th axis	5th axis
${\hat{\rho }}_{n}$	0.449	1.037	0.894	0.736	0.937
$T_{n}$	−20.398	−58.631	−46.649	−35.201	−49.741
	6th axis	7th axis	8th axis	9th axis	10th axis
${\hat{\rho }}_{n}$	0.951	0.971	1.017	0.958	0.967
$T_{n}$	−51.392	−52.607	−55.455	−51.221	−51.797
	11th axis	12th axis	13th axis	14th axis	15th axis
${\hat{\rho }}_{n}$	0.949	0.649	0.952	0.977	0.932
$T_{n}$	−50.457	−30.094	−50.633	−50.978	−48.842

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
