# Peer review of "Inference for Convolutionally Observed Diffusion Processes"

_entropy, 2020, doi:10.3390/e22091031_

Round 1
Reviewer 1 Report
This paper presents a method of observation of diffusion processes, based on a convolution technique.
As far as I can judge, the paper is well written, with objectives clearly stated, methods accurately defined, and sound conclusions. I suggest publication in its present form. A reference seems to be missing in the text (pages 2, 3, 12).
Author Response
We deeply appreciate your review and detailed comments on our manuscript.
We amended the problem of references you indicated.
Please check the new version of our manuscript.
We hope that the revision meets your approval.

Reviewer 2 Report
Journal: Entropy
Manuscript ID: entropy-923282
Title: Inference for convolutional observed diffusion processes
Authors: Shogo Nakakita *, Masayuki Uchida
The paper deals with a convolutional observation schemes; that is, the smoothness of observation against ordinary diffusion processes.
The presentation is accurate and the application of the theory is done on a linear additive SDE (the Ornstein-Uhlenbeck process). It would be nice if the author had some comments on a multiplicative process, which would have more interesting applications. For example the Kubo’s oscillator; i.e., the simplest 1D multiplicative SDE. Reference to this process can be found in many books on applied stochastic process and non-equilibrium statistics physics; see for example: Gardiner, van Kampen, Caceres, Risken, Horsthemke & Lefever, to mention a few of them).
In conclusion the paper is appropriate for publication after this point is amended.
Author Response
We deeply appreciate your review and detailed comments on our manuscript.
With respect to the multiplicative models, it is difficult to check whether they do satisfy moment and ergodicity conditions (for details, see [A1] and Remark 1 in Uchida and Yoshida, 2012).
Hence we have included the reference for sufficient conditions of our regularity conditions at the bottom of p.5.
We hope that the revision meets your approval.
